# MACHINE UNLEARNING IN LOW-DIMENSIONAL FEATURE SUBSPACE

## ABSTRACT

Data privacy in modern neural networks attracts growing interest in Machine Unlearning (MU), which aims at removing the knowledge of particular data from a pretrained model and meanwhile maintaining good performances on the remaining data. In this work, a new perspective upon low-dimensional feature subspaces is presented to investigate MU. We firstly demonstrate the potentials of separating the remaining and forgetting data in a low-dimensional feature subspace. Then, such separability motivates us to seek a subspace $\text{range}(\mathbf{U})$ on the features of the pretrained model for unlearning, where the information of the remaining data is preserved and that of the forgetting data is therein diminished, leading to the proposed new method named SUbspace UNlearning (SUN). Compared to mainstream MU methods that require direct and massive access to the training data for model updating, SUN offers two key advantages well resolving these significant challenges in practice. *(i)* SUN avoids frequent data visits and optimizes $\mathbf{U}$ involving two covariance matrices, which only requires one-shot feature fetching and thereby alleviates data privacy risks and computation. *(ii)* SUN in implementation simply serves as a plug-in module to the pretrained model without modifications to its original parameters, reducing the parameter number and computational overhead by orders of magnitude, which is of great practicality for handling multiple unlearning requests. Extensive numerical experiments verify our superior unlearning accuracy with significantly less parameters and computing time over variants of models, datasets, tasks, and applications. Code is available at the anonymous link https://anonymous.4open.science/r/4352/.

## 1 INTRODUCTION

Deep Neural Networks (DNNs) pretrained on large-scale data, e.g. web-crawled data, greatly advocate various applications (Schuhmann et al., 2022; Zha et al., 2025). However, it also raises significant privacy concerns due to its heavy reliance on massive data. Modern data regulatory frameworks, such as GDPR (Hoofnagle et al., 2019), emphasizes to preserve *the right to be forgotten* when requesting to delete particular data, e.g., private user data. To address this challenge, Machine Unlearning (MU) (Bourtoule et al., 2021) has emerged and attracted increasing interests in recent years. MU targets on dual goals, i.e., erasing the knowledge from the deleted (forgotten) data and meanwhile preserving the model performance. A gold-standard approach to MU is to retrain the DNN model from scratch based on the remaining data only, i.e., the entire training data excluding the forgetting data (Thudi et al., 2022b), which is called *the exact MU* (or the certificated MU). Such a retraining process is computationally expensive, especially for modern DNNs which are commonly large-scale models trained on massive data. Hence, a surge of researches on *approximate MU* (Izzo et al., 2021) has been developed to approach the ideal performances of the exact MU.

Existing approximate MU methods generally update the parameters of the original model pretrained on the entire training data $\mathcal{D}$, so as to align its outputs with that of the ideal model retrained in the exact MU. The core idea here is to proceed the parameter updates with an optimization objective that jointly penalizes the performances on the forgetting data $\mathcal{D}_{\text{fg}}$ and maintains the performances on the remaining data $\mathcal{D}_{\text{rm}}$, with $\mathcal{D}_{\text{fg}} \bigcup \mathcal{D}_{\text{rm}} = \mathcal{D}$ and $\mathcal{D}_{\text{fg}} \bigcap \mathcal{D}_{\text{rm}} = \emptyset$. These optimization objectives lead to different ways to modify the pretrained model, including fine-tuning only on $\mathcal{D}_{\text{rm}}$ (Warnecke et al., 2021), assigning random labels to $\mathcal{D}_{\text{fg}}$ (Golatkar et al., 2020), conducting gradient ascent on $\mathcal{D}_{\text{fg}}$ (Thudi et al., 2022a), masking those parameters sensitive to $\mathcal{D}_{\text{fg}}$ (Fan et al., 2024), distilling an additional network (Chundawat et al., 2023a; Zhou et al., 2025), etc. We provide detailed explana-

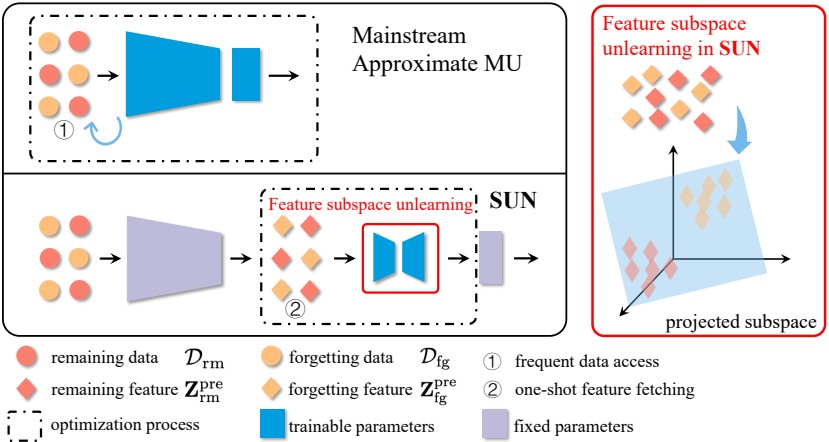

Figure 1: Overview of our proposed SUN and mainstream approximate MU methods. After unlearning, the trained projection matrix of SUN is incorporated into the model's forward propagation during inference, with details in Sec. 4.2.

tions and discussions on the related work in Appendix A. While enabling good performances, we note two critical drawbacks far underexplored in these MU methods:

• **Frequent visits to the massive training data.** Existing MU methods request to visit training samples in $\mathcal{D}_{\mathrm{rm}}$ and $\mathcal{D}_{\mathrm{fg}}$ during the iterative parameter updates to the pretrained model. Such massive data fetching not only incurs expensive computational overhead, but also poses privacy threats during the frequent access to (user) data.

• **Modification to the pretrained model.** Whenever a particular dataset $\mathcal{D}_{\mathrm{fg}}$ is requested to be forgotten, parameters of the pretrained model have to be updated, yielding a new model tailored to $\mathcal{D}_{\mathrm{fg}}$. This is inflexible and computationally expensive in practice, as each unlearning request demands that a new model be modified from the pretrained one.

In this work, we aim to address the above challenges in the approximate MU methods. *(i)* Firstly, a new perspective to study MU in low-dimensional feature subspaces is introduced. We reveal the potential separability in feature subspaces between $\mathcal{D}_{\mathrm{fg}}$ and $\mathcal{D}_{\mathrm{rm}}$ on the retrained model when comparing to the pretrained model. *(ii)* Then, the observed separability inspires us to learn a feature subspace from the pretrained model to achieve unlearning. To this end, the feature subspace is optimized to differentiate $\mathcal{D}_{\mathrm{fg}}$ and $\mathcal{D}_{\mathrm{rm}}$ by capturing maximal information of the features w.r.t. $\mathcal{D}_{\mathrm{rm}}$ and meanwhile describing minimal information w.r.t. $\mathcal{D}_{\mathrm{fg}}$. This optimization is implemented following the spirits of Principal Component Analysis (PCA) (Pearson, 1901): A projection matrix is trained and inserted into the pretrained model, requiring only two feature covariance matrices of $\mathcal{D}_{\mathrm{fg}}$ and $\mathcal{D}_{\mathrm{rm}}$. Consequently, the knowledge of $\mathcal{D}_{\mathrm{fg}}$ can be therein effectively diminished, while that of $\mathcal{D}_{\mathrm{rm}}$ gets well preserved in this feature subspace, thus fulfilling the objective of unlearning. Our method is thereby named as SUbspace UNlearning (SUN).

Fig. 1 provides an overview of our SUN and the related mainstream approximate MU methods. SUN offers efficiency for computation and storage, flexibility in practice, and privacy protection for user data, towards closing the gaps in adapting to real-world MU scenarios:

• **One-shot feature fetching**. SUN conducts MU in the feature space, and thus only needs to fetch the features outputted from the pretrained model, instead of directly and iteratively accessing training samples, e.g., private user data. Note that once the features are fetched, the calculation to its covariance matrix is one-shot, since our PCA-based optimization does not vary the covariance matrix during the iterative updates, but only optimizes projection directions for learning the feature subspace that well distinguishes the knowledge between $\mathcal{D}_{\mathrm{rm}}$ and $\mathcal{D}_{\mathrm{fg}}$. Hence, SUN successfully avoids direct and massive visits to the training data (user data) for privacy protection.

• **Plug-in module implementation.** With the spirits from PCA, we design a novel objective to optimize a projection matrix applied to the feature covariances, such that in the projected subspace the features w.r.t. $\mathcal{D}_{\mathrm{rm}}$ can be well reconstructed while the features w.r.t. $\mathcal{D}_{\mathrm{fg}}$ cannot. This algorithm enables the implementation in a plug-and-play manner: a projection matrix is inserted into the

pretrained model, without retraining or fine-tuning its model parameters. This merit is significant for efficiency and real-world practicality. For instance, when handling multiple unlearning requests from different users, SUN only requires a separate projection module in a rather small size for each request, instead of a new model modified from the pretrained DNN.

The key contributions of this work are summarized as follows:

- We presents a novel perspective, *feature subspace*, to investigate MU, which to the best of our knowledge is the first time revealing the potential separability between the forgetting and remaining data in low-dimensional feature subspaces.

- A new method named SUN is proposed by leveraging the subspace learning with PCA-based techniques, simply doing one-shot feature fetching and updating a projection matrix as a plugged-in module into the pretrained model. SUN well *addresses* the aforementioned crucial issues of *massive data access and pretrained model modification* in MU.

- The objective in SUN involves two covariance matrices with size depending on feature dimensions and only optimizes a projection matrix, *reducing the parameter number and computing time by orders of magnitude* than mainstream MU methods as shown in Table 2.

- Extensive experiments demonstrate our superior unlearning accuracy and efficiency on varied datasets, networks, unlearning tasks, and real-world applications.

## 2 PRELIMINARIES

**Machine Unlearning.** MU aims at maintaining prediction performances of a well-trained model $f_{\text{pre}}$ when removing (forgetting) specific data (Bourtoule et al., 2021). In MU, the model $f_{\text{pre}}$ pretrained on the full dataset $\mathcal{D}$ is given. A particular subset $\mathcal{D}_{\text{fg}}$ is requested to be forgotten, and we have the remaining data denoted as $\mathcal{D}_{\text{rm}} := \mathcal{D} \backslash \mathcal{D}_{\text{fg}}$. MU seeks to erase the knowledge of $\mathcal{D}_{\text{fg}}$ from $f_{\text{pre}}$ and meanwhile to preserve the performances on $\mathcal{D}_{\text{rm}}$. The golden standard in MU is to attain a model $f_{\text{exact}}$ retrained from scratch on $\mathcal{D}_{\text{rm}}$, i.e., the exact MU (Thudi et al., 2022b). In contrast, fine-tuning the pretrained model $f_{\text{pre}}$ to approximate the ideal performances of $f_{\text{exact}}$ has been a central research focus for flexibility and efficiency, namely the approximate MU (Izzo et al., 2021). We provide more discussions on related work in Appendix A.

**Principal Component Analysis.** PCA has long been being a fundamental tool in machine learning and beyond (Pearson, 1901; Abdi & Williams, 2010). PCA seeks orthogonal directions, i.e., principal components capturing the highest projection variance, so as to keep maximal data information. These directions can be obtained by taking the eigenvectors of the covariance matrix, whose size depends on data dimensions. PCA can explore the intrinsic patterns residing in low-dimension subspaces spanned by those principal components, and has also been studied for learning with modern DNNs, e.g., training algorithms (Li et al., 2023b) and anomaly detection applications (Huang et al., 2006; Fang et al., 2024). In this work, we leverage PCA schemes for MU to optimize the projection directions associated with the low-dimensional feature subspace of the pretrained model.

**Notations.** Let $f : \mathcal{X} \to \mathbb{R}^C$ be a DNN pretrained on $\mathcal{D} = \{\boldsymbol{x}_i, y_i\}_{i=1}^N \subseteq \mathcal{X} \times \mathcal{Y}$, with samples $\boldsymbol{x}_i \in \mathcal{X} \subset \mathbb{R}^D$ and their labels $y_i \in \mathcal{Y} = \{1, 2, \cdots, C\}$. The model $f(\cdot)$ takes $\boldsymbol{x}$ as the input and outputs $C$-dimensional logits $f(\boldsymbol{x}) \in \mathbb{R}^C$ for classification prediction. More specifically, $f$ can be structured with a backbone $g : \mathcal{X} \to \mathbb{R}^d$ and a linear layer $h : \mathbb{R}^d \to \mathbb{R}^C$ in sequence, where $g(\cdot)$ learns the (penultimate-layer) feature $\boldsymbol{z} = g(\boldsymbol{x}) \in \mathbb{R}^d$ and then the linear layer $h(\cdot)$ is applied to the feature $\boldsymbol{z}$ for the output logits, i.e., $f(\boldsymbol{x}) = h(g(\boldsymbol{x}))$.

## 3 FEATURE SUBSPACE PERSPECTIVE FOR APPROXIMATE MU

In this work, we introduce the perspective of feature subspaces to decouple the impacts of the forgetting data $\mathcal{D}_{\text{fg}}$ from the remaining data $\mathcal{D}_{\text{rm}}$ in approximate MU. This feature-based perspective helps explore the intrinsic differences between the pretrained $f_{\text{pre}}$ and the retrained $f_{\text{exact}}$ in low-dimensional subspaces of features, advocating efficient MU with private data protection.

The pretrained $f_{\text{pre}}$ is learned with the entire training dataset $\mathcal{D} = \mathcal{D}_{\text{fg}} \bigcup \mathcal{D}_{\text{rm}}$. In contrast, the ideal model $f_{\text{exact}}$ from the exact MU is trained exclusively on $\mathcal{D}_{\text{rm}}$ and never sees $\mathcal{D}_{\text{fg}}$. This fundamental distinction in their training paradigms motivates our exploration on the learned features $\boldsymbol{z}$ w.r.t. $\mathcal{D}_{\text{fg}}$ and $\mathcal{D}_{\text{rm}}$ from $f_{\text{pre}}$ and $f_{\text{exact}}$, respectively. To be specific, we consider the potential

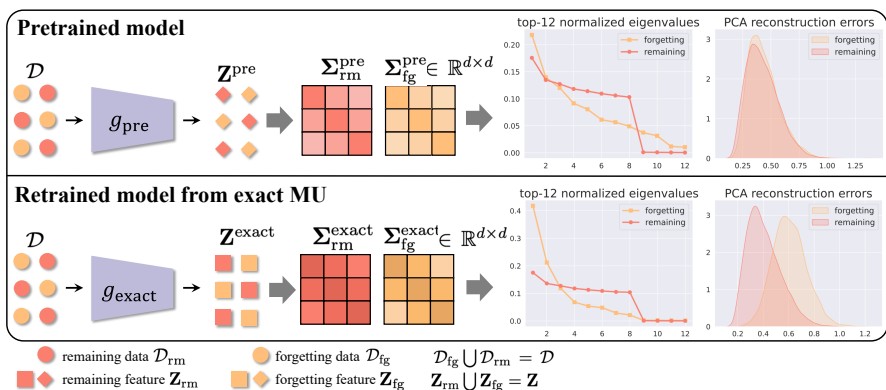

Figure 2: Illustration of spectrum analyses on the features of $\mathcal{D}_{\mathrm{fg}}$ and $\mathcal{D}_{\mathrm{rm}}$ w.r.t. the pretrained $f_{\mathrm{pre}}$ and the retrained $f_{\mathrm{exact}}$, with results on the top 12 normalized eigenvalues and reconstruction errors. Experiments are conducted on ResNet18 and CIFAR10, with one random class as $\mathcal{D}_{\mathrm{fg}}$.

separability between $\mathcal{D}_{\mathrm{fg}}$ and $\mathcal{D}_{\mathrm{rm}}$ through feature-based perspectives, naturally assuming that the unseen forgetting data $\mathcal{D}_{\mathrm{fg}}$ are easy to be differentiated from the seen $\mathcal{D}_{\mathrm{rm}}$ in the feature space learned in the exact MU model $f_{\mathrm{exact}}$. We in accordance make the following hypothesis:

(**H**) *For the exact MU model $f_{\mathrm{exact}}$, there exists a low-dimensional feature subspace, where the features of $\mathcal{D}_{\mathrm{fg}}$ and $\mathcal{D}_{\mathrm{rm}}$ are easy to be separated, while for the pretrained model $f_{\mathrm{pre}}$, the features of $\mathcal{D}_{\mathrm{fg}}$ and $\mathcal{D}_{\mathrm{rm}}$ remain distinctively more non-separable.*

We conduct spectrum analysis with PCA schemes to the features $z \in \mathbb{R}^d$ from the network backbone, in order to explore the intrinsic structures. We denote the features of the forgetting data $\mathcal{D}_{\mathrm{fg}}$ and the remaining data $\mathcal{D}_{\mathrm{rm}}$ from $g_{\mathrm{pre}}$ and $g_{\mathrm{exact}}$ as $\mathbf{Z}_{\mathrm{fg}}^{\mathrm{pre}}, \mathbf{Z}_{\mathrm{rm}}^{\mathrm{pre}}, \mathbf{Z}_{\mathrm{fg}}^{\mathrm{exact}}, \mathbf{Z}_{\mathrm{rm}}^{\mathrm{exact}}$ and their covariance matrices as $\mathbf{\Sigma}_{\mathrm{fg}}^{\mathrm{pre}}, \mathbf{\Sigma}_{\mathrm{rm}}^{\mathrm{pre}}, \mathbf{\Sigma}_{\mathrm{fg}}^{\mathrm{exact}}, \mathbf{\Sigma}_{\mathrm{rm}}^{\mathrm{exact}} \in \mathbb{R}^{d \times d}$, respectively. With eigen-decomposition to those covariance matrices, we investigate their eigenvalues and reconstruction performances.

**Spectrum analysis.** Eigen-decay indicates the relative importance across the corresponding principal components, i.e., the explained variance captured by such dimensions in the feature subspace. In this evaluation, we plot the results with top-12 components. In Fig. 2, $\mathbf{\Sigma}_{\mathrm{fg}}^{\mathrm{pre}}$ and $\mathbf{\Sigma}_{\mathrm{rm}}^{\mathrm{pre}}$ w.r.t. $\mathcal{D}_{\mathrm{fg}}$ and $\mathcal{D}_{\mathrm{rm}}$ from the pretrained model show similar trends in eigen-decay. For the retrained $f_{\mathrm{exact}}$, the eigen-decay of $\mathbf{\Sigma}_{\mathrm{fg}}^{\mathrm{exact}}$ is sharper than that of $\mathbf{\Sigma}_{\mathrm{rm}}^{\mathrm{exact}}$, which is different from the case with the vanilla pretrained model. This phenomenon reveals the potential of achieving unlearning in feature subspaces, rather than typically approaching the outputs or parameters of the exact retrained model in approximate machine unlearning methods.

**Reconstruction analysis.** The PCA reconstruction error quantitatively measures the captured information along projection directions. We apply PCA to the remaining features $\mathbf{Z}_{\mathrm{rm}}$ and obtain $\mathbf{U}_{\mathrm{rm}} \in \mathbb{R}^{d \times s}$ projecting the features onto the $s$-dimensional subspace. The reconstruction error for a centered feature $\bar{z}$ is calculated as $e = \|\mathbf{U}_{\mathrm{rm}}\mathbf{U}_{\mathrm{rm}}^{\top}\bar{z} - \bar{z}\|_2$. In Fig. 2, we compute such errors on both $\mathbf{Z}_{\mathrm{rm}}$ and $\mathbf{Z}_{\mathrm{fg}}$ of $f_{\mathrm{pre}}$ and $f_{\mathrm{exact}}$, respectively. The reconstruction errors for $\mathbf{Z}_{\mathrm{fg}}^{\mathrm{pre}}$ and $\mathbf{Z}_{\mathrm{rm}}^{\mathrm{pre}}$ of the pretrained $f_{\mathrm{pre}}$ are very similar, while those for $\mathbf{Z}_{\mathrm{fg}}^{\mathrm{exact}}$ and $\mathbf{Z}_{\mathrm{rm}}^{\mathrm{exact}}$ of the retrained $f_{\mathrm{exact}}$ appear distinctively different. The results suggest that the subspace learned from $\mathbf{Z}_{\mathrm{rm}}^{\mathrm{exact}}$ fails to capture the intrinsic patterns of $\mathbf{Z}_{\mathrm{fg}}^{\mathrm{exact}}$, making them easily separable therein and thus supporting the aforementioned hypothesis (**H**).

**Analytical evidence.** We provide theoretical evidence on reconstruction analyses for the hypothesis (**H**), supporting the promising perspective of learning a low-dimensional feature subspace that can separate $\mathcal{D}_{\mathrm{fg}}$ and $\mathcal{D}_{\mathrm{rm}}$ under the exact MU model $f_{\mathrm{exact}}$. This result is stated in the following Lemma 1 with proof in Appendix B.

**Lemma 1.** *Given the feature covariance $\mathbf{\Sigma}_{\mathrm{rm}}^{\mathrm{exact}}$ from $f_{\mathrm{exact}}$ on $\mathcal{D}_{\mathrm{rm}}$, its eigendecomposition is given by $\mathbf{\Sigma}_{\mathrm{rm}}^{\mathrm{exact}} = \mathbf{V}\mathbf{\Lambda}\mathbf{V}^{\top}$ with eigenvalues $\lambda_1 \geq \lambda_2 \geq \cdots \lambda_n$. There exists an $s$-dimensional subspace $\mathrm{range}(\mathbf{U}_*)$ spanned by the top-$s$ principal directions of $\mathbf{\Sigma}_{\mathrm{rm}}^{\mathrm{exact}}$ ($\mathbf{U}_* = \mathbf{V}_{:,1:s}$), such that for any $x \in \mathcal{D}_{\mathrm{rm}}$, its features can be well reconstructed by the subspace $\mathrm{range}(\mathbf{U}_*)$, while for any $x \in \mathcal{D}_{\mathrm{fg}}$,*

*its features hold minor projected information in* $\mathrm{range}(\mathbf{U}_*)$. *Mathematically, we have small positive numbers* $\epsilon_{\mathrm{rm}}, \epsilon_{\mathrm{fg}} > 0$ *that satisfies*

$$\|(\mathbf{I} - \mathbf{U}_*\mathbf{U}_*^\top)g_{\mathrm{exact}}(\boldsymbol{x})\|_2 \leq \epsilon_{\mathrm{rm}}, \quad \forall \boldsymbol{x} \in \mathcal{D}_{\mathrm{rm}},$$
$$\|\mathbf{U}_*\mathbf{U}_*^\top g_{\mathrm{exact}}(\boldsymbol{x})\|_2 \leq \epsilon_{\mathrm{fg}}, \quad \forall \boldsymbol{x} \in \mathcal{D}_{\mathrm{fg}}. \tag{1}$$

This insight unveils the distinction between $\mathbf{Z}_{\mathrm{rm}}$ and $\mathbf{Z}_{\mathrm{fg}}$ in low-dimensional feature subspaces w.r.t. $f_{\mathrm{exact}}$ and can advocate novel unlearning methodologies from the raised new perspective, such as our SUN introduced in Sec. 4.

## 4 APPROXIMATE MU WITH SUBSPACE LEARNING

Grounded on the perspective of feature subspaces and the analysis in Sec. 3, we now focus on how to attain a preferable feature subspace for $f_{\mathrm{pre}}$, such that the features $\mathbf{Z}_{\mathrm{rm}}^{\mathrm{pre}}$ and $\mathbf{Z}_{\mathrm{fg}}^{\mathrm{pre}}$ can be well distinguished in this subspace and the knowledge of the forgetting data $\mathcal{D}_{\mathrm{fg}}$ can be diminished from $f_{\mathrm{pre}}$. Accordingly, we name this proposed method as SUbspace UNlearning (SUN).

Note that the penultimate features applied with a linear last layer can lead to good classification accuracy, which is in fact done in the complete architecture of the pretrained model. This separability is on the labels for supervised classification tasks, but our addressed separability is between the remaining and the forgetting data (features) under general unlearning context. Our work investigates whether it is possible to formulate linear transformations (subspace learning) to these penultimate features, such that the knowledge of forgetting features can be diminished and that of remaining features is meanwhile well kept. Apart from approaching the outputs or parameters of the exact retrained model, it leads to an interesting and promising direction of formulating unlearning models that pertain similar properties in feature (sub)spaces, e.g., our proposed SUN, which we hope could bring new perspectives and opportunities in machine unlearning.

### 4.1 OPTIMIZATION

SUN proceeds to find an $s$-dimensional subspace $\mathrm{range}(\mathbf{U})$ for the $d$-dimensional features of the pretrained model $f_{\mathrm{pre}}$. This linear subspace is in fact given by the span of the orthonormal columns of the $d \times s$ matrix $\mathbf{U} = [\boldsymbol{u}_1, \ldots, \boldsymbol{u}_s]$, the set of which defines the well-known Stiefel manifold $\mathrm{St}(d, s)$ with $d \geq s$. Towards the feature separability for unlearning, SUN follows the spirits from PCA to seek such a subspace (to optimize $\mathbf{U}$), in which the information of $\mathbf{Z}_{\mathrm{rm}}^{\mathrm{pre}}$ is maximally captured and meanwhile that of the forgetting features $\mathbf{Z}_{\mathrm{fg}}^{\mathrm{pre}}$ gets minimally described.

**Proposition 1.** *Let* $\mathbf{M}$ *be an* $l \times l$ *symmetric matrix. Let* $\gamma_1, \ldots, \gamma_m$ *be its* $m$ *smallest eigenvalues, possibly including multiplicities, with associated orthonormal eigenvectors* $\boldsymbol{v}_1, \ldots, \boldsymbol{v}_m$. *Let* $\mathbf{V}$ *be a matrix whose columns are these eigenvectors. Then, the optimization problem* $\min_{\mathbf{U} \in \mathrm{St}(l,m)} \mathrm{Tr}(\mathbf{U}^\top \mathbf{M}\mathbf{U})$ *has a minimizer at* $\mathbf{U}_\star = \mathbf{V}$ *and we have* $\mathbf{U}_\star^\top \mathbf{M}\mathbf{U}_\star = \mathrm{diag}(\mathbf{\Gamma})$ *with* $\mathbf{\Gamma} = [\gamma_1, \ldots, \gamma_m]^\top$, *where* $\mathrm{Tr}(\cdot)$ *denotes the matrix trace.*

Proposition 1 (Pandey et al., 2022) formalizes the optimization problem for obtaining the projection directions capturing minimal information in PCA with Stiefel manifold, where $\mathbf{M}$ denotes the covariance matrix. This correspondingly leads to the optimization problem for the projection directions capturing maximal information: the subspace spanned by the eigenvectors of $\mathbf{M}$ with the $m$ largest eigenvalues is obtained by solving $\min_{\mathbf{U} \in \mathrm{St}(l,m)} \mathrm{Tr}(\mathbf{M} - \mathbf{U}\mathbf{U}^\top \mathbf{M}\mathbf{U}\mathbf{U}^\top)$, which also corresponds to the reconstruction error of PCA as similarly explained in Sec 4.1 of (Avron et al., 2014). In SUN, we establish the following objective for learning the subspace:

$$\min_{\mathbf{U} \in \mathrm{St}(d,s)} J(\mathbf{U}) = \underbrace{\left(\frac{\mathrm{Tr}(\mathbf{U}^\top \mathbf{\Sigma}_{\mathrm{fg}}^{\mathrm{pre}}\mathbf{U})}{\mathrm{Tr}(\mathbf{\Sigma}_{\mathrm{fg}}^{\mathrm{pre}})}\right)^2}_{J_{\mathrm{fg}}} + \underbrace{\left(\frac{\mathrm{Tr}\left(\mathbf{\Sigma}_{\mathrm{rm}}^{\mathrm{pre}} - \mathbf{U}\mathbf{U}^\top \mathbf{\Sigma}_{\mathrm{rm}}^{\mathrm{pre}}\mathbf{U}\mathbf{U}^\top\right)}{\mathrm{Tr}\left(\mathbf{\Sigma}_{\mathrm{rm}}^{\mathrm{pre}}\right)}\right)^2}_{J_{\mathrm{rm}}}, \tag{2}$$

where the numerators play key roles and the denominators balance normalizations. The patterns w.r.t. $\mathcal{D}_{\mathrm{rm}}$ get preserved with $J_{\mathrm{rm}}$, while that of $\mathcal{D}_{\mathrm{fg}}$ gets diminished with $J_{\mathrm{fg}}$. The joint optimization of $J_{\mathrm{rm}}$ and $J_{\mathrm{fg}}$ advocates the separability between $\mathbf{Z}_{\mathrm{rm}}^{\mathrm{pre}}$ and $\mathbf{Z}_{\mathrm{fg}}^{\mathrm{pre}}$ in the subspace projected via $\mathbf{U}$:

- $J_{\mathrm{fg}}$ quantifies the projection variance of the forgetting features $\mathbf{Z}_{\mathrm{fg}}^{\mathrm{pre}}$ onto the subspace spanned by $\mathbf{U}$, as explained in Proposition 1. Minimizing $J_{\mathrm{fg}}$ encourages $\mathbf{U}$ to capture minimal information of $\mathbf{Z}_{\mathrm{fg}}^{\mathrm{pre}}$, aiming to forget the knowledge of $\mathcal{D}_{\mathrm{fg}}$.

- $J_{\mathrm{rm}}$ measures the projection variance of the remaining features $\mathbf{Z}_{\mathrm{rm}}^{\mathrm{pre}}$ captured within the complement subspace w.r.t. $\mathbf{U}$, namely the reconstruction error. Minimizing $J_{\mathrm{rm}}$ forces $\mathbf{U}$ to preserve maximal information of $\mathbf{Z}_{\mathrm{rm}}^{\mathrm{pre}}$, so as to maintain model performances on $\mathcal{D}_{\mathrm{rm}}$.

## 4.2 IMPLEMENTATION AND DISCUSSION

SUN simply requires to optimize the projection matrix $\mathbf{U}$ applied to the features of the pretrained model $f_{\mathrm{pre}}$, which in this work are primarily chosen from the penultimate layer (the backbone $g_{\mathrm{pre}}$). In implementation, a projection module associated with parameters $\mathbf{U}$ is inserted between the backbone $g_{\mathrm{pre}}$ and the last linear layer $h_{\mathrm{pre}}$. The idea behind is to encode data into a subspace of the feature space, where the orthogonal projector onto `range(U)` is $\mathbf{U}\mathbf{U}^\top$. We deploy Riemannian optimizer (Becigneul & Ganea, 2019; Kochurov et al., 2020) for $J(\mathbf{U})$ with parameters optimized on Stiefel manifold. The resulting model from SUN for unlearning is then given by $f_{\mathbf{U}}(\cdot) \triangleq h_{\mathrm{pre}}(\mathbf{U}\mathbf{U}^\top g_{\mathrm{pre}}(\cdot))$, in contrast to the original pretrained model $f_{\mathrm{pre}} = h_{\mathrm{pre}}(g_{\mathrm{pre}}(\cdot))$.

**One-shot Feature Fetching.**  The computation to $J(\mathbf{U})$ relies on the two covariance matrices $\mathbf{\Sigma}_{\mathrm{rm}}^{\mathrm{pre}}$ and $\mathbf{\Sigma}_{\mathrm{fg}}^{\mathrm{pre}}$ w.r.t. the features from the pretrained model $f_{\mathrm{pre}}$. Once the covariances $\mathbf{\Sigma}_{\mathrm{rm}}^{\mathrm{pre}}, \mathbf{\Sigma}_{\mathrm{fg}}^{\mathrm{pre}}$ are computed, SUN proceeds to optimize the $d \times s$ projection matrix $\mathbf{U}$ and thus only conducts one-shot feature fetching. In the existing approximate MU methods, the original training data are visited in each iteration to update their models. This characteristic of our SUN avoids direct and massive visits to data samples, which is not only efficient in data accessing but also important for privacy protection on (user) data in real-world MU.

**Plug-in Module Implementation.**  Compared to fine-tuning or retraining the entire parameters in $f_{\mathrm{pre}}$, SUN optimizes a $d \times s$ matrix as in (2), the parameter number of which is thereby reduced by orders of magnitude. Thus, SUN takes significantly less computation overhead than the exact MU and mainstream approximate MU methods, with numerical supports of Table 2 in Sec. 5.2. In implementation, SUN flexibly serves as a plug-in module into the pretrained model $f_{\mathrm{pre}}$, without modifying $f_{\mathrm{pre}}$. This merit is of great practicality in real-world deployment: in cases of handling multiple unlearning requests from different users, each request can be addressed by efficiently optimizing a small-size projection matrix, instead of maintaining multiple models refined from $f_{\mathrm{pre}}$.

**Theoretical guarantee.**  A series of researches (Guo et al., 2020; Zhang et al., 2024) in certified unlearning provide theoretical guarantees on the approximation performance between their unlearned models and the exact MU model by leveraging the differential privacy framework (Dwork, 2006). Critically, these works offer certification from a parameter-based perspective, i.e., bounding the difference between the parameters of the unlearned model and the exact MU model, since unlearning is executed by updating parameters of the pretrained model $f_{\mathrm{pre}}$. In contrast, SUN employs a fundamentally different mechanism for unlearning: SUN leaves the pretrained model parameters unchanged, and instead inserts a learnable projection matrix $\mathbf{U}$ into $f_{\mathrm{pre}}$. Unlearning is then accomplished by modifying the intermediate features from $f_{\mathrm{pre}}$ according to the objective in (2), ensuring the final model outputs $f_{\mathbf{U}}(\cdot)$ contain minimal knowledge of $\mathcal{D}_{\mathrm{fg}}$. In this sense, the existing parameter-based theoretical framework of certified unlearning is not applicable to SUN for approximation analysis. We then shed light on the *output* differences between SUN ($f_{\mathbf{U}}$) and the exact MU model ($f_{\mathrm{exact}}$) to establish a theoretical guarantee, as formalized in Theorem 1 with proofs in Appendix B.

**Theorem 1.** *Suppose that $J(\mathbf{U})$ is optimized with a stationary point $\hat{\mathbf{U}}$ that approximates the optimal solution $\mathbf{U}_*$ with a small gap $\epsilon_{\mathrm{opt}} > 0$, such that $\|\mathbf{U}_*\mathbf{U}_*^\top - \hat{\mathbf{U}}\hat{\mathbf{U}}^\top\|_F \leq \epsilon_{\mathrm{opt}}$, then, for any $\boldsymbol{x} \in \mathcal{D}$, the difference of outputs between the unlearned model $f_{\hat{\mathbf{U}}}$ and the exact MU model $f_{\mathrm{exact}}$ is bounded by a constant $C$: $\|f_{\hat{\mathbf{U}}}(\boldsymbol{x}) - f_{\mathrm{exact}}(\boldsymbol{x})\|_2 \leq C$ with $C$ dependent on $\epsilon_{\mathrm{opt}}$.*

## 5 EXPERIMENTS

In this section, extensive experiments are conducted to evaluate SUN and investigations on practical applications and ablation studies are also provided, showing potentials of SUN and its perspective of low-dimensional feature subspaces for MU. In experiments, we primarily focus on the class-centric unlearning, widely studied in real-world scenarios (Zhou et al., 2025), as data from different users can represent varied classes, where tasks of extreme and continual unlearning are further explored. The instance-wise unlearning is also investigated, which is another significant unlearning type. We provide setup details in Appendix C and additional experiment results and discussions in Sec. D.

Table 1: Comparison results of Swin-T on Tiny-ImageNet. The ✓ and ✗ indicate whether $\mathcal{D}_{rm}$ and $\mathcal{D}_{fg}$ are involved in optimization. Avg.G. is average accuracy and MIA gap to the retrained model.

| method | $\mathcal{D}_{rm}$ | $\mathcal{D}_{fg}$ | $Acc_{rm}^{tr}$ | $Acc_{fg}^{tr}$ | $Acc_{rm}^{te}$ | $Acc_{fg}^{te}$ | MIA | Avg.G.↓ |
|---|---|---|---|---|---|---|---|---|
| pretrained | ✓ | ✓ | $99.63_{\pm0.01}$ | $99.75_{\pm0.20}$ | $74.54_{\pm0.16}$ | $74.83_{\pm8.14}$ | $96.63_{\pm1.02}$ | - |
| retrained | ✓ | ✗ | $99.67_{\pm0.03}$ (0.00) | $0.00_{\pm0.00}$ (0.00) | $75.44_{\pm0.19}$ (0.00) | $0.00_{\pm0.00}$ (0.00) | $0.00_{\pm0.00}$ (0.00) | 0.00 |
| GA | ✗ | ✓ | $88.40_{\pm5.51}$ (11.27) | $23.12_{\pm12.14}$ (23.12) | $65.27_{\pm3.56}$ (10.17) | $18.82_{\pm7.52}$ (18.82) | $19.95_{\pm10.24}$ (19.95) | 16.67 |
| FT | ✓ | ✗ | $99.92_{\pm0.00}$ (0.25) | $18.73_{\pm5.90}$ (18.73) | $74.25_{\pm0.21}$ (1.19) | $16.17_{\pm6.33}$ (16.17) | $3.73_{\pm2.22}$ (3.73) | 8.02 |
| RL | ✗ | ✓ | $92.94_{\pm1.91}$ (6.73) | $9.98_{\pm7.44}$ (9.98) | $67.17_{\pm1.46}$ (8.17) | $7.50_{\pm7.57}$ (7.50) | $1.83_{\pm0.54}$ (1.83) | 6.85 |
| RL | ✓ | ✓ | $99.91_{\pm0.01}$ (0.24) | $1.52_{\pm1.17}$ (1.52) | $74.43_{\pm0.18}$ (1.01) | $0.33_{\pm0.58}$ (0.33) | $0.00_{\pm0.00}$ (0.00) | 0.62 |
| SalUn | ✗ | ✓ | $93.16_{\pm1.90}$ (6.51) | $13.55_{\pm10.30}$ (13.55) | $67.49_{\pm1.56}$ (7.95) | $10.00_{\pm9.26}$ (10.00) | $3.45_{\pm0.87}$ (3.45) | 8.29 |
| SalUn | ✓ | ✓ | $99.88_{\pm0.01}$ (0.21) | $2.57_{\pm1.09}$ (2.57) | $74.71_{\pm0.19}$ (0.73) | $1.00_{\pm0.87}$ (1.00) | $0.00_{\pm0.00}$ (0.00) | 0.90 |
| BT | ✗ | ✓ | $91.70_{\pm1.89}$ (7.97) | $7.72_{\pm5.45}$ (7.72) | $66.35_{\pm1.37}$ (9.09) | $5.50_{\pm6.14}$ (5.50) | $1.40_{\pm0.44}$ (1.40) | 6.34 |
| L2UL | ✗ | ✓ | $92.69_{\pm3.43}$ (6.98) | $4.43_{\pm0.15}$ (4.43) | $67.69_{\pm2.58}$ (7.75) | $2.00_{\pm0.87}$ (2.00) | $2.57_{\pm0.92}$ (2.57) | 4.75 |
| DELETE | ✗ | ✓ | $99.13_{\pm0.12}$ (**0.54**) | $13.32_{\pm4.17}$ (13.32) | $73.31_{\pm0.28}$ (2.13) | $6.50_{\pm2.18}$ (6.50) | $1.85_{\pm0.48}$ (1.85) | 4.88 |
| **SUN** | only $\Sigma_{rm}^{pre}$ | only $\Sigma_{fg}^{pre}$ | $98.59_{\pm0.07}$ (1.08) | $1.48_{\pm1.27}$ (**1.48**) | $74.27_{\pm0.18}$ (**1.17**) | $0.50_{\pm0.50}$ (**0.50**) | $0.00_{\pm0.00}$ (**0.00**) | **0.85** |

Table 2: Comparison on computational costs of Swin-T on Tiny-ImageNet w.r.t. the results in Table 1. RTE is the run time efficiency in seconds on single NVIDIA GeForce RTX 4090 GPU.

| method | $\mathcal{D}_{rm}$ | $\mathcal{D}_{fg}$ | # optimized param. | RTE (seconds)↓ | additional requirements |
|---|---|---|---|---|---|
| pretrained | ✓ | ✓ | 27,673,154 | 882.05 | - |
| retrain | ✓ | ✗ | 27,673,154 | 858.14 | - |
| GA | ✗ | ✓ | 27,673,154 | 13.49 | - |
| FT | ✓ | ✗ | 27,673,154 | 428.24 | - |
| RL | ✗ | ✓ | 27,673,154 | 13.19 | - |
| RL | ✓ | ✓ | 27,673,154 | 438.48 | - |
| SalUn | ✗ | ✓ | 13,836,577 | 15.00 | a parameter saliency mask |
| SalUn | ✓ | ✓ | 13,836,577 | 450.23 | a parameter saliency mask |
| BT | ✗ | ✓ | 27,673,154 | 14.32 | a randomly-initialized teacher model |
| L2UL | ✗ | ✓ | 27,673,154 | 15225.05 | a set of adversarial examples; a parameter importance mask |
| DELETE | ✗ | ✓ | 27,673,154 | 22.14 | a copy of the pretrained model |
| **SUN** | only $\Sigma_{rm}^{pre}$ | only $\Sigma_{fg}^{pre}$ | **192,000** | **0.56** | a projection matrix $\mathbf{U} \in \text{St}(768, 250)$ |

## 5.1 SETUPS

**Datasets and models.** Experiments are evaluated on Tiny-ImageNet (Le & Yang, 2015) and ImageNet-1K (Deng et al., 2009) datasets, where $64 \times 64 \times 3$ and $224 \times 224 \times 3$ images are categorized into 200 and 1,000 classes, respectively. We skip comparisons on other easier datasets with smaller scales and distinctively less classes, e.g., CIFAR10 (Krizhevsky, 2009) and SVHN (Netzer et al., 2011), which have been extensively evaluated and yet showed almost incremental improvements with limited challenges for class-centric unlearning. For Tiny-ImageNet and ImageNet-1K, we evaluate on Swin-T (Liu et al., 2021) and ResNet50 (He et al., 2016), respectively. More structures such as ResNet18 (He et al., 2016) are also discussed in experiments.

**Baselines.** We compare SUN with a series of strong baselines: Fine-Tuning (FT) (Warnecke et al., 2021), Gradient Ascent (GA) (Thudi et al., 2022a), Random Labeling (RL) (Golatkar et al., 2020), Salun (Fan et al., 2024), Bad Teaching (BT) (Chundawat et al., 2023a), Learn to UnLearn (L2UL) (Cha et al., 2024), and DELETE (Zhou et al., 2025). Different from those methods, our method does not directly access to the training samples, but only conducts one-shot feature-fetching from the pretrained model. Following recent studies that emphasize MU only using the forgetting data $\mathcal{D}_{fg}$ for practicality in real-world setups (Cha et al., 2024; Zhou et al., 2025), our comparison results are presented among those baselines operating on $\mathcal{D}_{fg}$. Methods involving $\mathcal{D}_{rm}$ are also provided for reference and marked in gray. The performances of the pretrained model $f_{pre}$ on the full training set $\mathcal{D}$ and the exact MU model $f_{exact}$ retrained on the remaining data $\mathcal{D}_{rm}$ are presented as well.

**Metrics.** Four typical evaluations metrics on learning accuracy are leveraged (Zhou et al., 2025): $Acc_{fg}^{tr}$ on the forgetting $\mathcal{D}_{fg}$ and $Acc_{rm}^{tr}$ on the remaining $\mathcal{D}_{rm}$ of the training data, $Acc_{fg}^{te}$ on the forgetting $\mathcal{D}_{fg}^{te}$ and $Acc_{rm}^{te}$ on the remaining $\mathcal{D}_{rm}^{te}$ of the test data. In addition, the Membership Inference Attack success rate MIA (Shokri et al., 2017) is adopted, measuring the proportion of samples in $\mathcal{D}_{fg}$ that gets memorized by the unlearned model. We mark the best results highlighted in bold font. Note that the performance gap to the exact retrained $f_{exact}$ essentially reflects the unlearning efficacy of approximate MU methods. This is the key evaluation criterion for MU and is marked in blue within parentheses. Hence, the overall performances are indicated by the average of the accuracy gap and the MIA gap to $f_{exact}$ (Avg.G.).

Table 3: Comparison results under single-class unlearning with ResNet50 on ImageNet-1K.

| method | $\mathcal{D}_{rm}$ | $\mathcal{D}_{fg}$ | $Acc_{rm}^{tr}$ | $Acc_{fg}^{tr}$ | $Acc_{rm}^{te}$ | $Acc_{fg}^{te}$ | Avg.G.↓ | RTE↓ | # parameters |
|---|---|---|---|---|---|---|---|---|---|
| pretrained | ✓ | ✓ | $77.92_{\pm0.01}$ | $83.46_{\pm7.64}$ | $76.15_{\pm0.01}$ | $81.33_{\pm10.26}$ | - | - | 25,557,032 |
| retrained | ✓ | ✗ | $80.25_{\pm0.16}$ (0.00) | $0.00_{\pm0.00}$ (0.00) | $75.72_{\pm0.04}$ (0.00) | $0.00_{\pm0.00}$ (0.00) | 0.00 | - | 25,557,032 |
| GA | ✗ | ✓ | $71.87_{\pm1.80}$ (8.38) | $0.00_{\pm0.00}$ (0.00) | $71.42_{\pm1.89}$ (4.30) | $0.00_{\pm0.00}$ (0.00) | 3.17 | 9.32 | 25,557,032 |
| RL | ✗ | ✓ | $72.11_{\pm1.42}$ (8.14) | $1.87_{\pm0.69}$ (1.87) | $71.53_{\pm1.44}$ (4.19) | $5.33_{\pm4.16}$ (5.33) | 4.88 | 14.40 | 25,557,032 |
| SalUn | ✗ | ✓ | $71.56_{\pm0.68}$ (8.69) | $2.67_{\pm0.77}$ (2.67) | $70.84_{\pm0.60}$ (4.88) | $4.67_{\pm3.06}$ (4.67) | 5.23 | 27.73 | 12,778,516 |
| BT | ✗ | ✓ | $72.07_{\pm1.53}$ (8.18) | $1.87_{\pm0.93}$ (1.87) | $71.49_{\pm1.49}$ (4.23) | $5.33_{\pm4.16}$ (5.33) | 4.90 | 16.84 | 25,557,032 |
| DELETE | ✗ | ✓ | $72.46_{\pm1.30}$ (7.79) | $0.95_{\pm0.78}$ (0.95) | $71.90_{\pm1.23}$ (**3.82**) | $2.00_{\pm3.46}$ (2.00) | 3.64 | 17.82 | 25,557,032 |
| SUN | only $\Sigma_{rm}^{pre}$ | only $\Sigma_{fg}^{pre}$ | $79.96_{\pm0.38}$ (**0.29**) | $0.00_{\pm0.00}$ (**0.00**) | $69.40_{\pm0.35}$ (6.32) | $0.00_{\pm0.00}$ (**0.00**) | **1.65** | **0.94** | **1,024,000** |

Table 4: Comparison on performance under extreme unlearning with Swin-T on Tiny-ImageNet.

| method | $\mathcal{D}_{rm}$ | $\mathcal{D}_{fg}$ | $Acc_{rm}^{tr}$ | $Acc_{fg}^{tr}$ | $Acc_{rm}^{te}$ | $Acc_{fg}^{te}$ | MIA | Avg.G.↓ | RTE↓ | # parameters |
|---|---|---|---|---|---|---|---|---|---|---|
| pretrained | ✓ | ✓ | $99.64_{\pm0.02}$ | $99.63_{\pm0.00}$ | $75.43_{\pm2.61}$ | $74.45_{\pm0.29}$ | $95.34_{\pm0.43}$ | - | 882.05 | 27,673,154 |
| retrained | ✓ | ✗ | $99.97_{\pm0.01}$ (0.00) | $0.00_{\pm0.00}$ (0.00) | $90.57_{\pm0.75}$ (0.00) | $0.00_{\pm0.00}$ (0.00) | $0.00_{\pm0.00}$ (0.00) | 0.00 | 95.17 | 27,673,154 |
| GA | ✗ | ✓ | $14.26_{\pm6.06}$ (85.71) | $9.61_{\pm4.36}$ (9.61) | $12.20_{\pm4.61}$ (78.37) | $8.10_{\pm3.46}$ (8.10) | $11.41_{\pm5.19}$ (11.41) | 38.64 | 178.07 | 27,673,154 |
| FT | ✓ | ✗ | $100.00_{\pm0.01}$ (0.03) | $69.98_{\pm1.24}$ (69.98) | $91.13_{\pm0.86}$ (0.56) | $49.98_{\pm1.03}$ (49.98) | $38.82_{\pm0.58}$ (38.82) | 31.88 | 47.42 | 27,673,154 |
| RL | ✗ | ✓ | $26.98_{\pm3.11}$ (72.99) | $16.33_{\pm2.69}$ (16.33) | $23.43_{\pm2.05}$ (67.14) | $13.75_{\pm2.35}$ (13.75) | $5.39_{\pm0.75}$ (5.39) | 35.12 | 195.26 | 27,673,154 |
| RL | ✓ | ✓ | $98.77_{\pm0.22}$ (1.20) | $20.40_{\pm0.83}$ (20.40) | $90.03_{\pm0.58}$ (0.54) | $15.38_{\pm0.47}$ (15.38) | $1.10_{\pm0.51}$ (1.10) | 7.72 | 431.45 | 27,673,154 |
| SalUn | ✗ | ✓ | $27.39_{\pm0.48}$ (72.58) | $17.57_{\pm3.28}$ (17.57) | $23.30_{\pm2.1}$ (67.27) | $15.05_{\pm2.65}$ (15.05) | $12.25_{\pm1.57}$ (12.25) | 36.95 | 235.23 | 27,673,154 |
| SalUn | ✓ | ✓ | $98.72_{\pm0.37}$ (1.25) | $13.88_{\pm3.09}$ (13.88) | $87.93_{\pm1.33}$ (2.64) | $11.61_{\pm2.29}$ (11.61) | $2.74_{\pm1.14}$ (2.14) | 6.31 | 476.78 | 27,673,154 |
| BT | ✗ | ✓ | $24.77_{\pm1.45}$ (75.20) | $14.66_{\pm1.40}$ (14.66) | $21.30_{\pm1.32}$ (69.27) | $12.57_{\pm1.30}$ (12.57) | $4.09_{\pm0.44}$ (4.09) | 35.16 | 88.81 | 27,673,154 |
| L2UL | ✗ | ✓ | $23.90_{\pm7.94}$ (76.07) | $15.00_{\pm3.50}$ (15.00) | $19.10_{\pm6.35}$ (71.47) | $11.91_{\pm2.50}$ (11.91) | $3.66_{\pm1.95}$ (3.66) | 35.63 | 7361.76 | 27,673,154 |
| DELETE | ✗ | ✓ | $97.37_{\pm0.36}$ (2.60) | $6.95_{\pm0.31}$ (6.95) | $81.07_{\pm1.53}$ (9.50) | $10.96_{\pm0.48}$ (10.96) | $1.16_{\pm0.16}$ (1.16) | 6.23 | 446.62 | 27,673,154 |
| SUN | only $\Sigma_{rm}^{pre}$ | only $\Sigma_{fg}^{pre}$ | $99.63_{\pm0.18}$ (**0.34**) | $0.03_{\pm0.03}$ (**0.03**) | $92.17_{\pm0.65}$ (**1.60**) | $0.02_{\pm0.02}$ (**0.02**) | $0.08_{\pm0.12}$ (**0.08**) | **0.42** | **0.29** | **13,824** |

## 5.2 MAIN COMPARISONS

**Multi/Single-class unlearning.** Tables 1 and 2 present comparisons on the unlearning results and computational costs with Swin-T on Tiny-ImageNet, where 4 of 200 classes are randomly selected as $\mathcal{D}_{fg}$. Table 1 shows that unlearning without access to $\mathcal{D}_{rm}$ leads to substantial performance drops for baselines such as RL and SalUn, showing their strong reliance to $\mathcal{D}_{rm}$. Our SUN leverages the two covariance matrices of features and achieves state-of-the-art unlearning performances with a smallest average gap of 0.85 to the retrained model. On efficiency, Table 2 shows that the parameter number and the Run Time Efficiency (RTE) in our SUN are less by orders of magnitude than that of other baselines, as our subspace learning simply optimizes a projection matrix operating on two covariance matrices and others generally update the pretrained model with involving the original training data. For single-class unlearning, we experiment on the much more challenging ImageNet-1K dataset with ResNet50, where 1 of 1000 classes is randomly selected as $\mathcal{D}_{fg}$ with 3 repeated runs. Given the large scale of this dataset ($\mathcal{D}_{rm}$ contains over 1.28M samples), we therefore focus on approximate MU methods that only utilize $\mathcal{D}_{fg}$, ensuring computation efficiency. The results in Table 3 demonstrates that SUN outperforms other baselines with the smallest average gap, together with the shortest running time (less than 1 second) and the fewest number of parameters (reduced by approximately 95%).

**Extreme unlearning.** We conduct evaluations to investigate the robustness under extreme unlearning scenarios, where a very high ratio (over 90%) of the training samples is removed, i.e., 180 of 200 classes in Tiny-ImageNet are randomly selected as $\mathcal{D}_{fg}$ with results in Table 4. In this extreme unlearning scenario, methods that directly update $f_{pre}$ struggle to maintain good performances on $\mathcal{D}_{rm}$, as the gradient updates to $f_{pre}$ are dominated by reducing performances on $\mathcal{D}_{fg}$ that contains a substantial number of samples. In contrast, our SUN does not modify $f_{pre}$ and its projection matrix is optimized on the feature covariances w.r.t. $\mathcal{D}_{rm}$ and $\mathcal{D}_{fg}$, which is less affected by the relative data sizes of $\mathcal{D}_{rm}$ and $\mathcal{D}_{fg}$. SUN achieves the best average gap (Avg.G.) of 0.42 with about 0.05% parameters w.r.t. the given model, and takes the shortest running time (RTE) about 0.29 seconds.

**Instance unlearning.** SUN can also be applied to instance-wise unlearning with results shown in Table 5, where 1,000 of 100,000 training samples (1%) in Tiny-ImageNet are randomly selected as $\mathcal{D}_{fg}$. Accuracy on the whole test dataset is evaluated as $Acc^{te}$. Under this random forgetting setting, $\mathcal{D}_{fg}$ and $\mathcal{D}_{rm}$ can contain samples from all classes, which is more challenging to find a feature subspace well differentiating $\mathcal{D}_{fg}$ and $\mathcal{D}_{rm}$. Nevertheless, in Table 5, the low MIA gap of SUN implies that its learned subspace can effectively remove information related to $\mathcal{D}_{fg}$. Furthermore, SUN attains superior performance and efficiency on the overall Avg.G. metric across all baselines by optimizing only about 1% parameters w.r.t. the given model in 0.5 second.

**Extended discussions.** We further investigate MU tasks in real-world applications, including *face recognition* and *emotion recognition*, the former of which is conducted on VGGFace2 (Cao et al.,

Table 5: Comparison results under instance unlearning with Swin-T on Tiny-ImageNet.

| method | $\mathcal{D}_{rm}$ | $\mathcal{D}_{fg}$ | $Acc_{rm}^{tr}$ | $Acc_{fg}^{tr}$ | $Acc^{te}$ | MIA | Avg.G.↓ | RTE ↓ | # parameters |
|---|---|---|---|---|---|---|---|---|---|
| pretrained | ✓ | ✓ | $99.63_{\pm0.00}$ | $99.80_{\pm0.01}$ | $74.55_{\pm0.00}$ | $96.20_{\pm0.44}$ | - | 882.05 | 27,673,154 |
| retrained | ✓ | ✗ | $99.67_{\pm0.02}$ (0.00) | $75.97_{\pm1.00}$ (0.00) | $75.03_{\pm0.10}$ (0.00) | $64.47_{\pm1.25}$ (0.00) | 0.00 | 857.21 | 27,673,154 |
| GA | ✗ | ✓ | $98.37_{\pm0.53}$ (1.30) | $93.63_{\pm0.46}$ (17.66) | $72.36_{\pm0.55}$ (**2.67**) | $85.77_{\pm1.10}$ (21.30) | 10.73 | 7.92 | 27,673,154 |
| RL | ✗ | ✓ | $97.56_{\pm0.62}$ (2.11) | $95.00_{\pm1.64}$ (19.03) | $70.57_{\pm0.50}$ (4.46) | $88.83_{\pm0.99}$ (24.36) | 12.49 | 7.61 | 27,673,154 |
| SalUn | ✗ | ✓ | $98.47_{\pm0.34}$ (**1.20**) | $97.23_{\pm1.01}$ (21.26) | $71.86_{\pm0.24}$ (3.17) | $89.60_{\pm1.08}$ (25.13) | 12.69 | 12.79 | 13,836,577 |
| BT | ✗ | ✓ | $96.46_{\pm0.44}$ (3.21) | $92.83_{\pm0.92}$ (16.86) | $69.52_{\pm0.35}$ (5.51) | $85.23_{\pm0.78}$ (20.76) | 11.59 | 8.23 | 27,673,154 |
| L2UL | ✗ | ✓ | $94.77_{\pm0.51}$ (4.90) | $74.90_{\pm0.10}$ (**1.07**) | $68.69_{\pm0.27}$ (6.34) | $73.90_{\pm0.46}$ (9.43) | 5.44 | 7919.11 | 27,673,154 |
| DELETE | ✗ | ✓ | $91.84_{\pm2.12}$ (7.83) | $77.57_{\pm2.39}$ (1.60) | $65.44_{\pm1.43}$ (9.59) | $68.67_{\pm1.50}$ (4.20) | 5.81 | 8.42 | 27,673,154 |
| **SUN** | only $\Sigma_{rm}^{pre}$ | only $\Sigma_{fg}^{pre}$ | $92.65_{\pm0.54}$ (7.02) | $82.97_{\pm1.17}$ (7.00) | $68.14_{\pm0.42}$ (6.89) | $64.47_{\pm2.36}$ (**0.00**) | **5.23** | **0.50** | **230,400** |

Table 6: Comparison results of ResNet50 on VGGFace2, forgetting one of 200 identities.

| method | $\mathcal{D}_{rm}$ | $\mathcal{D}_{fg}$ | $Acc_{rm}^{tr}$ | $Acc_{fg}^{tr}$ | $Acc_{rm}^{te}$ | $Acc_{fg}^{te}$ | MIA | Avg.G.↓ | RTE ↓ | # parameters |
|---|---|---|---|---|---|---|---|---|---|---|
| pretrained | ✓ | ✓ | 100.00 | 100.00 | 98.15 | 99.49 | 100.00 | - | 5970.95 | 25,557,032 |
| retrained | ✓ | ✗ | 100.00 (0.00) | 0.00 (0.00) | 98.21 (0.00) | 0.00 (0.00) | 0.00 (0.00) | 0.00 | 5870.35 | 25,557,032 |
| GA | ✗ | ✓ | 91.91 (8.09) | 0.00 (0.00) | 86.34 (11.87) | 0.00 (0.00) | 0.00 (0.00) | 4.00 | 3.35 | 25,557,032 |
| FT | ✓ | ✗ | 99.84 (0.16) | 0.00 (0.00) | 96.22 (1.99) | 0.00 (0.00) | 0.00 (0.00) | 0.43 | 294.62 | 25,557,032 |
| RL | ✗ | ✓ | 99.64 (0.36) | 11.25 (11.25) | 95.27 (2.94) | 13.20 (13.20) | 0.00 (0.00) | 5.55 | 5.59 | 25,557,032 |
| RL | ✓ | ✗ | 99.11 (0.89) | 0.00 (0.00) | 94.66 (3.55) | 0.00 (0.00) | 0.00 (0.00) | 0.89 | 303.81 | 25,557,032 |
| SalUn | ✗ | ✓ | 99.68 (**0.32**) | 5.50 (5.50) | 95.59 (**2.62**) | 5.08 (5.08) | 0.00 (0.00) | 2.70 | 9.99 | 12,778,516 |
| SalUn | ✓ | ✗ | 99.96 (0.04) | 0.00 (0.00) | 97.00 (1.21) | 0.00 (0.00) | 0.00 (0.00) | 0.25 | 312.71 | 12,778,516 |
| BT | ✗ | ✓ | 99.39 (0.61) | 5.00 (5.00) | 94.97 (3.24) | 4.06 (4.06) | 0.00 (0.00) | 2.58 | 8.24 | 25,557,032 |
| L2UL | ✗ | ✓ | 98.09 (1.91) | 2.75 (2.75) | 92.88 (5.33) | 3.55 (3.55) | 0.00 (0.00) | 2.71 | 176.69 | 25,557,032 |
| DELETE | ✗ | ✓ | 99.50 (0.50) | 5.50 (5.50) | 95.15 (3.06) | 5.58 (5.58) | 0.00 (0.00) | 2.93 | 5.82 | 25,557,032 |
| **SUN** | only $\Sigma_{rm}^{pre}$ | only $\Sigma_{fg}^{pre}$ | 97.20 (2.80) | 0.00 (**0.00**) | 94.22 (3.99) | 0.51 (**0.51**) | 0.00 (**0.00**) | **1.46** | **0.45** | **43,008** |

2018) and ResNet50 and the latter of which is done on RAF-DB (Li et al., 2017) and ResNet18. Both identities and emotions are key facial attributes closely related to user privacy, exploring the unlearning of such information can benefit privacy protection. Table 6 demonstrates our advantageous performances in face recognition, with more results and setup details left to Appendix E. Due to space limitation, another practical unlearning scenario of multiple unlearning requests in sequence, i.e., *continual unlearning*, is discussed with experiments provided in Appendix D, further verifying the effectiveness of our SUN after multiple rounds of continual unlearning.

### 5.3 ABLATION STUDY AND SENSITIVITY ANALYSIS

**Objective terms $J_{rm}$ and $J_{fg}$.** We conduct experiments to explore the roles of the two loss terms in the optimization objective $J(\mathbf{U})$ of (2). These two terms together aim at the dual goals in MU: $J_{rm}$ preserves the performances on $\mathcal{D}_{rm}$ and $J_{fg}$ guarantees the unlearning on $\mathcal{D}_{fg}$. By respectively removing $J_{rm}$ and $J_{fg}$ from $J(\mathbf{U})$, we report the performances in Table 7. It shows that

Table 7: Ablation study on $J_{rm}$ and $J_{fg}$ with Swin-T and Tiny-ImageNet.

| objective | $Acc_{rm}^{tr}$ | $Acc_{fg}^{tr}$ | $Acc_{rm}^{te}$ | $Acc_{fg}^{te}$ | MIA |
|---|---|---|---|---|---|
| None | 97.11 | 98.10 | 71.61 | 65.00 | 86.10 |
| $J_{fg}$ only | 89.51 | 11.50 | 65.40 | 4.00 | 0.00 |
| $J_{rm}$ only | 99.19 | 87.80 | 75.07 | 53.00 | 58.35 |
| $J_{rm} + J_{fg}$ | 98.66 | 1.10 | 74.27 | 0.00 | 0.00 |

optimizing $\mathbf{U}$ only with $J_{fg}$ suppresses $Acc_{fg}$ and MIA, while failing to preserve $Acc_{rm}$. Employing only $J_{rm}$ manages to achieve high $Acc_{rm}$, but leaves $\mathcal{D}_{fg}$ hardly forgotten with high $Acc_{fg}$ and MIA. Hence, a joint optimization with both $J_{rm}$ and $J_{fg}$ is crucial for MU.

**Subspace dimensions $s$.** The projection matrix $\mathbf{U}$ is confined on Stiefel manifold $St(d, s)$, consisting of $s$ orthonormal bases in columns. The hyper-parameter $s$ indicates the dimension of the projected subspace and therein affects the captured information. A sensitivity analysis w.r.t. varied values of $s$ on the unlearning performance is provided in the left panel of Fig. 3. Recall Fig. 2 that the number of components (dimension $s$) determines the preserved information (explained variance in PCA) from features. As illustrated in Fig. 3, a larger value of $s$ leads to smaller $Acc_{rm}$ together with more computations, and vice versa. From our evaluations, we recommend initializing $s$ with preserving 95% of the explained variance in $\Sigma_{rm}^{pre}$, as this choice readily provides good empirical performances for different settings and tasks evaluated in this work.

**Implementation position for subspace learning.** The subspace learning in our SUN is primarily applied to the penultimate-layer features $z$, where the projection matrix $\mathbf{U}$ is inserted between $g_{pre}$ and $h_{pre}$. In practice, the implementation of SUN is quite flexible to be inserted into different positions in $f_{pre}$. Here, we implement SUN as a plug-in module in different layers with $f_{pre}$ exemplified on ResNet18, which is structured with a preceding module followed by 4 `layer` modules and a linear layer module.[1] Fig. 3 presents the ablation results. When SUN is applied in earlier positions,

---

[1] https://docs.pytorch.org/vision/main/_modules/torchvision/models/resnet.html#resnet18

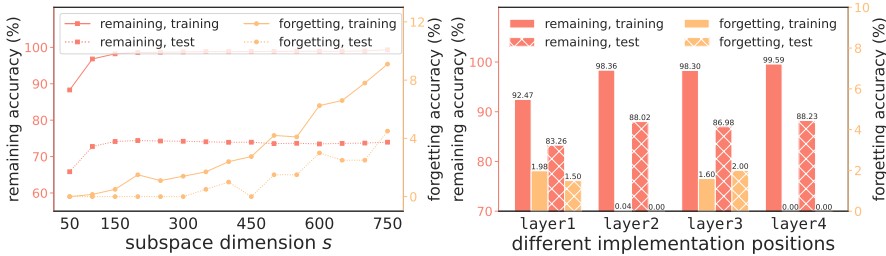

Figure 3: A sensitivity analysis on *(i)* subspace dimensions $s$ (**left**) and *(ii)* different positions of the projection matrix $\mathbf{U}$ (**right**). For *(i)*: experiments are conducted on Swin-T with Tiny-ImageNet where $d = 768$ and 4 classes are randomly selected as $\mathcal{D}_{\text{fg}}$. For *(ii)*: experiments are on ResNet18 with CIFAR10 dataset, where one class is randomly selected as $\mathcal{D}_{\text{fg}}$.

competitive performances can be maintained. These results demonstrate the flexibility of SUN and indicate that the covariance matrices in earlier layers contain effective information for achieving unlearning. However, it can cost more computations, due to the larger feature dimensions and more optimization steps with early-layer implementation, as given by Table A6 in Appendix E.1. Considering the trade-off between performance and efficiency, it is recommended to implement SUN in the penultimate layer as done in main evaluations.

## 6 CONCLUSION AND DISCUSSION

This work presents the perspective of low-dimensional feature subspaces to investigate MU, which is by far underexplored and shows promising potentials with our evaluations. Our key insight lies in the potential separability between the remaining and forgetting data in feature subspaces and seeks to learn such a subspace for the pretrained model. On such bases, we propose a novel method named SUN that deploys PCA-based techniques to learn a projection matrix associated with the desired low-dimensional feature subspace, in which the knowledge of the remaining data is preserved and that of the forgetting data is minimally remained. In optimization, only one-shot feature fetching is required to compute two covariances, avoiding direct and massive visits to the original training data for privacy protection. SUN simply updates a small-size projection matrix and in implementation serves as a plug-in module to the pretrained model without modifying its original parameters, which is of great practicality for handling multiple unlearning requests. This efficient training paradigm of SUN reduces both the parameter number and the running time by orders of magnitude, while achieving superior unlearning accuracy as supported by our extensive numerical evaluations.

SUN is currently applied to discriminative models for classification, and not yet extended to generation, as generative models are more susceptible to the internal changes in feature layers. It would be interesting to further investigate the subspace unlearning for image generation in future work.

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

# APPENDIX

## A RELATED WORK ON MACHINE UNLEARNING

Machine unlearning aims at maintaining prediction performances of a well-trained model $f_{\text{pre}}$ when removing (forgetting) specific data (Cao & Yang, 2015; Ginart et al., 2019; Bourtoule et al., 2021). In general, such forgetting data $\mathcal{D}_{\text{fg}}$ can mainly be categorized into class-wise and instance-wise types. The former indicates that $\mathcal{D}_{\text{fg}}$ includes all the samples from one or multiple classes in $\mathcal{Y}$ (Tarun et al., 2023; Chundawat et al., 2023b; Zhou et al., 2025) while the latter implies that $\mathcal{D}_{\text{fg}}$ can be a random subset of the entire training data $\mathcal{D}$ with mixed classes and even all classes (Kim & Woo, 2022; Fan et al., 2024; Cha et al., 2024).

A golden standard of MU is to retrain the model from scratch on the remaining data $\mathcal{D}_{\text{rm}}$ only, known as *the exact MU* (Bourtoule et al., 2021; Thudi et al., 2022b). The resulting retrained model $f_{\text{exact}}$ never sees the forgetting data $\mathcal{D}_{\text{fg}}$ and thereby is an oracle in evaluating MU performances. However, the retraining of the given model on $\mathcal{D}_{\text{rm}}$ can suffer from heavy computational overload, which motivates a series of researches optimizing the model by approaching to the ideal performances of the exacted MU, namely *the approximated MU*. Existing methods in approximate MU generally update the parameters of the pretrained model $f_{\text{pre}}$ via different techniques, usually requiring to visit the forgetting data $\mathcal{D}_{\text{fg}}$ and/or the remaining data $\mathcal{D}_{\text{rm}}$ from the original training dataset. In the following, the involved MU methods in our comparisons are outlined, covering mainstream unlearning techniques and marking their reliance to $\mathcal{D}_{\text{fg}}$ and $\mathcal{D}_{\text{rm}}$.

**FT** (Golatkar et al., 2020; Warnecke et al., 2021) fine-tunes the pretrained model $f_{\text{pre}}$ only on the remaining data $\mathcal{D}_{\text{rm}}$ via minimizing the cross-entropy loss exemplified on the classification task. In most cases, $\mathcal{D}_{\text{fg}}$ only accounts for a small proportion of the entire training set, and thereby the time consuming for FT per epoch remains relatively high.

**GA** (Graves et al., 2021; Thudi et al., 2022a) fine-tunes $f_{\text{pre}}$ by applying gradient ascent only on the forgetting data $\mathcal{D}_{\text{fg}}$, implemented by maximizing the cross entropy loss. GA is highly sensitive to the learning rate and usually leads to a substantial drop on the performance of $\mathcal{D}_{\text{rm}}$.

**RL** (Golatkar et al., 2020) fine-tunes $f_{\text{pre}}$ by replacing the labels of $\mathcal{D}_{\text{fg}}$ with new different labels. RL is implemented with both $\mathcal{D}_{\text{fg}}$ and $\mathcal{D}_{\text{rm}}$ involved and can easily approach the performance of the ideal retrained model $f_{\text{exact}}$. Nevertheless, the existing researches have highlighted the importance of MU with only $\mathcal{D}_{\text{fg}}$ accessible (Cha et al., 2024; Zhou et al., 2025). Under this more practical setting with limited data access, RL excluding $\mathcal{D}_{\text{rm}}$ in optimization shows unsatisfactory accuracy on $\mathcal{D}_{\text{rm}}$ and $\mathcal{D}_{\text{fg}}$, verified by our empirical results in this work.

**SalUn** (Fan et al., 2024) proposes to mask those parameters in $f_{\text{pre}}$ that are sensitive to $\mathcal{D}_{\text{fg}}$. In implementation, a parameter saliency mask is derived based on $\mathcal{D}_{\text{fg}}$ only, and then is applied to the fine-tuning of $f_{\text{pre}}$. The fine-tuning of SalUn follows RL and similarly faces the aforementioned challenge as RL. SalUn excluding $\mathcal{D}_{\text{rm}}$ in the fine-tuning shows uncompetitive unlearning results and the saliency mask does not bring performance gains to RL when $\mathcal{D}_{\text{fg}}$ is only available. Another similar technique is to leverage parameter sparsity by weight pruning (Jia et al., 2023). Besides SalUn, SFR-on (Huang et al., 2024) further introduces gradient information via Hessians into the parameter mask and proposes a unified optimization framework with a fast-slow weight update. SFR-on also heavily relies on $\mathcal{D}_{\text{rm}}$ in computing the Fisher information matrix and the loss function.

**BT** (Chundawat et al., 2023a) introduces knowledge distillation for approximated MU. There are two teacher models in BT: an incompetent teacher model being a randomly-initialized network and a competent teacher model being the pretrained model $f_{\text{pre}}$. The student model is also the pretrained network $f_{\text{pre}}$. During unlearning, the competent (incompetent) teacher distills information from $\mathcal{D}_{\text{rm}}$ ($\mathcal{D}_{\text{fg}}$) to the student via the KL divergence. The competent teacher and the student actually share the same network in memory, implying that the competent teacher gets updated during distillation. Although both $\mathcal{D}_{\text{rm}}$ and $\mathcal{D}_{\text{fg}}$ are involved, our experiments show that BT solely with $\mathcal{D}_{\text{fg}}$, i.e., a single incompetent teacher, still achieves good results. SCRUB (Kurmanji et al., 2023) is also based on distillation, with the pretrained model as the teacher. The pretrained model is optimized under the guidance of the teacher by minimizing (maximizing) the KL divergence with $\mathcal{D}_{\text{rm}}$ ($\mathcal{D}_{\text{fg}}$) and a cross-entropy loss on $\mathcal{D}_{\text{rm}}$.

**L2UL** (Cha et al., 2024) introduces adversarial examples (AEs) (Madry et al., 2018) and weight importance (Aljundi et al., 2018) to promote MU given access only to $\mathcal{D}_{\text{fg}}$. As claimed in L2UL, AEs w.r.t. $\mathcal{D}_{\text{fg}}$ can mimic the remaining data $\mathcal{D}_{\text{rm}}$, so that minimizing the cross-entropy loss on

those AEs can benefit the performance on $\mathcal{D}_{\mathrm{rm}}$. A parameter importance mask is further applied, where changes of weights that are less important in classifying $\mathcal{D}_{\mathrm{fg}}$ get penalized. L2UL is particularly superior in instance-wise unlearning. However, the computation burden in generating AEs is considerably heavy, e.g., the default setup of 100 attack iterations and 200 AEs per image for ImageNet-1K in L2UL.

**DELETE** (Zhou et al., 2025) is a distillation-based method relying only on $\mathcal{D}_{\mathrm{fg}}$. In DELETE, a copy of the pretrained network is utilized as the teacher model, and the key is to apply a mask to the outputs from the teacher. This mask aims at altering the predictive probabilities of $\mathcal{D}_{\mathrm{fg}}$ to approximate the ideal case and is implemented in a simple way: positions in the mask w.r.t. the ground-true labels of $\mathcal{D}_{\mathrm{fg}}$ are set to negative infinity and others are set to zero.

Beyond the baselines previously discussed, we further review several subspace-based MU methods that are more closely relevant with our SUN. These methods typically define their subspaces in a non-learnable way based on network parameters. Gradient subspaces in convolution neural networks are identified in (Li et al., 2023a; Fu et al., 2024) through eigen-decomposition. In (Lizzo & Heck, 2025), the subspace in large language models is constructed by applying the Gram-Schmidt process to the singular vectors of parameters. In contrast, our SUN fundamentally differs from those works by leveraging a learnable feature subspace for unlearning.

Researches in MU have been extended to a wide ranges of fields with different tasks. The classwise and concept forgetting in image generation is explored in (Fan et al., 2024; Huang et al., 2024). Privacy attacks against MU methods and corresponding defenses are discussed in (Xiao et al., 2025). Unlearning from adversarial trained models is exploited to achieve superior unlearning performances and also strong adversarial robustness (Liu et al., 2023). The concept of unnecessary unlearning is formalized in (Li et al., 2025) with an algorithm to filter those unnecessary requests. MU can be an effective tool for defending the backdoor attacks (Liu et al., 2022; 2024) through forgetting the backdoor triggers hidden in data. Moreover, MU in large language models has also received significant attention with extensive researches in recent years (Xu et al., 2025; Liu et al., 2025).

# B  PROOFS

## B.1  PROOF OF LEMMA 1

*Proof.* The ideal unlearning model $f_{\mathrm{exact}}$ is trained solely on the remaining data $\mathcal{D}_{\mathrm{rm}}$, such that $f_{\mathrm{exact}}$ sufficiently learns knowledge from $\mathcal{D}_{\mathrm{rm}}$ and has never seen the forgetting data $\mathcal{D}_{\mathrm{fg}}$. We thereby assume that the remaining features $\mathbf{Z}_{\mathrm{fg}}^{\mathrm{exact}}$ learned by the backbone of $f_{\mathrm{exact}}$ satisfy that the variance of $\mathbf{Z}_{\mathrm{rm}}^{\mathrm{exact}}$ is distributed compactly. That is, for the eigenvalues $\lambda_1 \geq \lambda_2 \geq \cdots \lambda_d > 0$ of $\mathbf{\Sigma}_{\mathrm{rm}}^{\mathrm{exact}} \in \mathbb{R}^{d \times d}$, $\exists$ a small positive integer $s$ such that $s \ll d$, we have $\frac{\sum_{i=1}^{s} \lambda_i}{\sum_{i=1}^{d} \lambda_i} \geq 1 - \xi$, where $\xi > 0$ is a very small positive number, e.g. $\xi = 0.05$.

For any feature $\boldsymbol{z}_{\mathrm{rm}}^{\mathrm{exact}}$ of $\mathcal{D}_{\mathrm{rm}}$ from $g_{\mathrm{exact}}$, we have:

$$\mathbb{E}\left[\|(\mathbf{I} - \mathbf{U}_* \mathbf{U}_*^\top)(\boldsymbol{z}_{\mathrm{rm}}^{\mathrm{exact}} - \boldsymbol{\mu}_{\mathrm{rm}}^{\mathrm{exact}})\|_2^2\right] = \sum_{i=s+1}^{d} \lambda_i = \xi \cdot \mathrm{Tr}(\mathbf{\Sigma}_{\mathrm{rm}}^{\mathrm{exact}}), \tag{3}$$

where $\boldsymbol{\mu}_{\mathrm{rm}}^{\mathrm{exact}}$ denotes the means of $\mathbf{Z}_{\mathrm{rm}}^{\mathrm{exact}}$. Therefore, $\|(\mathbf{I} - \mathbf{U}_* \mathbf{U}_*^\top)g_{\mathrm{exact}}(\boldsymbol{x})\|_2 \leq \epsilon_{\mathrm{rm}}$ for $\boldsymbol{x} \in \mathcal{D}_{\mathrm{rm}}$ holds with $\epsilon_{\mathrm{rm}} \propto \sqrt{\xi}$.

Similarly, for any features $\boldsymbol{z}_{\mathrm{fg}}^{\mathrm{exact}}$ of $\mathcal{D}_{\mathrm{fg}}$ from $g_{\mathrm{exact}}$, we have:

$$\begin{aligned}
\|\mathbf{U}_* \mathbf{U}_*^\top \boldsymbol{z}_{\mathrm{fg}}^{\mathrm{exact}}\|_2^2 &= \|\mathbf{U}_* \mathbf{U}_*^\top (\boldsymbol{\mu}_{\mathrm{fg}}^{\mathrm{exact}} + \boldsymbol{z}_{\mathrm{fg}}^{\mathrm{exact}} - \boldsymbol{\mu}_{\mathrm{fg}}^{\mathrm{exact}})\|_2^2 \\
&\leq \left(\|\mathbf{U}_* \mathbf{U}_*^\top \boldsymbol{\mu}_{\mathrm{fg}}^{\mathrm{exact}}\|_2 + \|\mathbf{U}_* \mathbf{U}_*^\top (\boldsymbol{z}_{\mathrm{fg}}^{\mathrm{exact}} - \boldsymbol{\mu}_{\mathrm{fg}}^{\mathrm{exact}})\|_2\right)^2,
\end{aligned} \tag{4}$$

where we can assume the boundedness of feature means: $\|\boldsymbol{\mu}_{\mathrm{fg}}^{\mathrm{exact}}\|_2 \leq M_{\boldsymbol{\mu}}$. Since $\mathbf{U}_*$ is derived from $\mathcal{D}_{\mathrm{rm}}$, and $\boldsymbol{\mu}_{\mathrm{fg}}^{\mathrm{exact}}$ comes from data that are not involved during the training of $f_{\mathrm{exact}}$, we introduce an coefficient $\alpha$ to measure the alignment between $\boldsymbol{\mu}_{\mathrm{fg}}^{\mathrm{exact}}$ and the subspace $\mathbf{U}_*$: $\|\mathbf{U}_* \mathbf{U}_*^\top \boldsymbol{\mu}_{\mathrm{fg}}^{\mathrm{exact}}\|_2 \leq \alpha \cdot \|\boldsymbol{\mu}_{\mathrm{fg}}^{\mathrm{exact}}\|_2 \leq \alpha_{\max} \cdot M_{\boldsymbol{\mu}}$ with an upper bound $\alpha_{\max}$ for $\alpha$.

Besides, the expectation of $\|\mathbf{U}_*\mathbf{U}_*^\top(\boldsymbol{z}_{\text{fg}}^{\text{exact}} - \boldsymbol{\mu}_{\text{fg}}^{\text{exact}})\|_2$ is given by

$$
\begin{aligned}
\mathbb{E}\left[\|\mathbf{U}_*\mathbf{U}_*^\top(\boldsymbol{z}_{\text{fg}}^{\text{exact}} - \boldsymbol{\mu}_{\text{fg}}^{\text{exact}})\|_2^2\right] &= \mathbb{E}\left[(\boldsymbol{z}_{\text{fg}}^{\text{exact}} - \boldsymbol{\mu}_{\text{fg}}^{\text{exact}})^\top \mathbf{U}_*\mathbf{U}_*^\top(\boldsymbol{z}_{\text{fg}}^{\text{exact}} - \boldsymbol{\mu}_{\text{fg}}^{\text{exact}})\right] \\
&= \text{Tr}(\mathbf{U}_*^\top \boldsymbol{\Sigma}_{\text{fg}}^{\text{exact}} \mathbf{U}_*).
\end{aligned}
\tag{5}
$$

We denote $\sigma_{\max}^2$ to bound the maximal projection variance: $\|\boldsymbol{\Sigma}_{\text{fg}}^{\text{exact}\,1/2}\mathbf{U}_*\|_2^2 \le \sigma_{\max}^2$, and then we have $\|\mathbf{U}_*\mathbf{U}_*^\top(\boldsymbol{z}_{\text{fg}}^{\text{exact}} - \boldsymbol{\mu}_{\text{fg}}^{\text{exact}})\|_2 \le \sqrt{s}\cdot\sigma_{\max} + \mathcal{O}(1)$. Therefore, $\forall \boldsymbol{x} \in \mathcal{D}_{\text{fg}}$, $\|\mathbf{U}_*\mathbf{U}_*^\top g_{\text{exact}}(\boldsymbol{x})\|_2 \le \epsilon_{\text{fg}}$ holds for $\epsilon_{\text{fg}} = \alpha_{\max}\cdot M_{\boldsymbol{\mu}} + \sqrt{s}\cdot\sigma_{\max} + \mathcal{O}(1)$. The proof finishes. $\qquad\square$

**Remark 1.** *We particularly discuss the scenarios when $\mathcal{D}_{\text{fg}}$ and $\mathcal{D}_{\text{rm}}$ share high similarities. In this case, the exact MU model $f_{\text{exact}}$, though trained on $\mathcal{D}_{\text{rm}}$ only, inevitably encode partial knowledge of $\mathcal{D}_{\text{fg}}$. Correspondingly, in our proof above, the projection matrix $\mathbf{U}_*$ from the eigendecomposition on $\mathbf{Z}_{\text{rm}}^{\text{exact}}$ also contains variance of $\mathbf{Z}_{\text{fg}}^{\text{exact}}$, resulting in larger upper bounds $\alpha_{\max}$ and $\sigma_{\max}$ for the alignment coefficient $\alpha$ and the projection variance $\|\boldsymbol{\Sigma}_{\text{fg}}^{\text{exact}\,1/2}\mathbf{U}_*\|_2^2$. Then, the bound $\epsilon_{\text{fg}}$ for $\|\mathbf{U}_*\mathbf{U}_*^\top g_{\text{exact}}(\boldsymbol{x})\|_2$ becomes even be loose, and the separability of $\mathbf{U}_*$ might be less effective.*

### B.2 PROOF OF THEOREM 1

We firstly present the assumptions required for the proof of Theorem 1.

**Assumption 1.** *The pretrained network $f_{\text{pre}}$ and the retrained network $f_{\text{exact}}$ differ in whether the forgetting data are involved into training, and show nearly the same performance on the remaining data. In this sense, for any $\boldsymbol{x} \in \mathcal{D}$, we can assume that differences in their learned features are bounded: $\|g_{\text{pre}}(\boldsymbol{x}) - g_{\text{exact}}(\boldsymbol{x})\|_2 \le \epsilon_d$ and the output features of $f_{\text{exact}}$ are assumed to be bounded as $\|g_{\text{exact}}(\boldsymbol{x})\|_2 \le \epsilon_{\text{exact}}$.*

**Assumption 2.** *The last linear layer $h_{\text{pre}}(\cdot)$ in the pretrained network $f_{\text{pre}}$ is $L_{\text{pre}}$-Lipschitz continuous:*

$$
\|h_{\text{pre}}(\boldsymbol{z}_1) - h_{\text{pre}}(\boldsymbol{z}_2)\|_2 \le L_{\text{pre}} \cdot \|\boldsymbol{z}_1 - \boldsymbol{z}_2\|_2, \quad \forall \boldsymbol{z}_1, \boldsymbol{z}_2 \in \mathbb{R}^d.
\tag{6}
$$

*Specifically, the linear layer $h_{\text{pre}}(\cdot)$ with parameters $\mathbf{W}_{\text{pre}}$, i.e., $h_{\text{pre}}(\boldsymbol{z}) = \mathbf{W}_{\text{pre}}^\top \boldsymbol{z}$, has the Lipschitz constant $L_{\text{pre}} = \|\mathbf{W}_{\text{pre}}\|_2$. Similarly, the linear layer $h_{\text{exact}}$ in the exact MU model $f_{\text{exact}}$ has the Lipschitz constant $L_{\text{exact}} = \|\mathbf{W}_{\text{exact}}\|_2$ with its parameters $\mathbf{W}_{\text{exact}}$. We assume that the difference between parameters $\mathbf{W}_{\text{pre}}$ and $\mathbf{W}_{\text{exact}}$ is bounded by $\|\mathbf{W}_{\text{pre}} - \mathbf{W}_{\text{exact}}\|_2 \le \epsilon_W$.*

The proof of Theorem 1 is given below.

*Proof.* We consider two cases that the input $\boldsymbol{x}$ is from $\mathcal{D}_{\text{rm}}$ and $\mathcal{D}_{\text{fg}}$, respectively. For any data $\boldsymbol{x} \in \mathcal{D}_{\text{rm}}$, we have:

$$
\begin{aligned}
\|f_{\hat{\mathbf{U}}}(\boldsymbol{x}) - f_{\text{exact}}(\boldsymbol{x})\|_2 \le &\|h_{\text{pre}}(\hat{\mathbf{U}}\hat{\mathbf{U}}^\top g_{\text{pre}}(\boldsymbol{x})) - h_{\text{pre}}(\hat{\mathbf{U}}\hat{\mathbf{U}}^\top g_{\text{exact}}(\boldsymbol{x}))\|_2 \\
&+ \|h_{\text{pre}}(\hat{\mathbf{U}}\hat{\mathbf{U}}^\top g_{\text{exact}}(\boldsymbol{x})) - h_{\text{pre}}(g_{\text{exact}}(\boldsymbol{x}))\|_2 \\
&+ \|h_{\text{pre}}(g_{\text{exact}}(\boldsymbol{x})) - h_{\text{exact}}(g_{\text{exact}}(\boldsymbol{x}))\|_2.
\end{aligned}
\tag{7}
$$

We denote the 3 terms in Eqn.(7) as follows:

$$
\begin{aligned}
A &= \|h_{\text{pre}}(\hat{\mathbf{U}}\hat{\mathbf{U}}^\top g_{\text{pre}}(\boldsymbol{x})) - h_{\text{pre}}(\hat{\mathbf{U}}\hat{\mathbf{U}}^\top g_{\text{exact}}(\boldsymbol{x}))\|_2, \\
B &= \|h_{\text{pre}}(\hat{\mathbf{U}}\hat{\mathbf{U}}^\top g_{\text{exact}}(\boldsymbol{x})) - h_{\text{pre}}(g_{\text{exact}}(\boldsymbol{x}))\|_2, \\
C &= \|h_{\text{pre}}(g_{\text{exact}}(\boldsymbol{x})) - h_{\text{exact}}(g_{\text{exact}}(\boldsymbol{x}))\|_2.
\end{aligned}
\tag{8}
$$

These 3 terms $A$, $B$ and $C$ are bounded, respectively as follows:

$$
\begin{aligned}
A =&\|h_{\text{pre}}(\hat{\mathbf{U}}\hat{\mathbf{U}}^\top g_{\text{pre}}(\boldsymbol{x})) - h_{\text{pre}}(\hat{\mathbf{U}}\hat{\mathbf{U}}^\top g_{\text{exact}}(\boldsymbol{x}))\|_2 \\
\le& L_{\text{pre}} \cdot \|\hat{\mathbf{U}}\hat{\mathbf{U}}^\top(g_{\text{pre}}(\boldsymbol{x}) - g_{\text{exact}}(\boldsymbol{x}))\|_2 \quad \text{(Assumption 2)} \\
\le& L_{\text{pre}} \cdot \|\hat{\mathbf{U}}\hat{\mathbf{U}}^\top\|_2 \cdot \|g_{\text{pre}}(\boldsymbol{x}) - g_{\text{exact}}(\boldsymbol{x})\|_2 \\
\le& L_{\text{pre}} \cdot \epsilon_d, \quad \text{(Assumption 1)}
\end{aligned}
\tag{9}
$$

$$\begin{aligned} B =& \|h_{\mathrm{pre}}(\hat{\mathbf{U}}\hat{\mathbf{U}}^\top g_{\mathrm{exact}}(\boldsymbol{x})) - h_{\mathrm{pre}}(g_{\mathrm{exact}}(\boldsymbol{x}))\|_2 \\ \leq& L_{\mathrm{pre}} \cdot \|(\mathbf{I} - \hat{\mathbf{U}}\hat{\mathbf{U}}^\top)g_{\mathrm{exact}}(\boldsymbol{x})\|_2 \quad \text{(Assumption 2)} \\ \leq& L_{\mathrm{pre}} \cdot (\|(\mathbf{I} - \mathbf{U}_*\mathbf{U}_*^\top)g_{\mathrm{exact}}(\boldsymbol{x})\|_2 + \|(\mathbf{U}_*\mathbf{U}_*^\top - \hat{\mathbf{U}}\hat{\mathbf{U}}^\top)g_{\mathrm{exact}}(\boldsymbol{x})\|_2) \\ \leq& L_{\mathrm{pre}} \cdot (\epsilon_{\mathrm{rm}} + \|\mathbf{U}_*\mathbf{U}_*^\top - \hat{\mathbf{U}}\hat{\mathbf{U}}^\top\|_F \cdot \|g_{\mathrm{exact}}(\boldsymbol{x})\|_2) \quad \text{(Lemma 1)} \\ \leq& L_{\mathrm{pre}} \cdot (\epsilon_{\mathrm{rm}} + \mathcal{O}(\epsilon_{\mathrm{opt}})), \end{aligned} \tag{10}$$

where $\epsilon_{\mathrm{rm}}$ is from the Lemma 1, and $C = \|h_{\mathrm{pre}}(g_{\mathrm{exact}}(\boldsymbol{x})) - h_{\mathrm{exact}}(g_{\mathrm{exact}}(\boldsymbol{x}))\|_2 \leq \|\mathbf{W}_{\mathrm{pre}} - \mathbf{W}_{\mathrm{exact}}\|_2 \cdot \|g_{\mathrm{exact}}(\boldsymbol{x})\|_2 \leq \epsilon_W \cdot \epsilon_{\mathrm{exact}}$ according to Assumption 2.

Similarly, for any data $\boldsymbol{x} \in \mathcal{D}_{\mathrm{fg}}$, we we the decomposition on $\|f_{\hat{\mathbf{U}}}(\boldsymbol{x}) - f_{\mathrm{exact}}(\boldsymbol{x})\|_2$, such that

$$\begin{aligned} \|f_{\hat{\mathbf{U}}}(\boldsymbol{x}) - f_{\mathrm{exact}}(\boldsymbol{x})\|_2 \leq& \|h_{\mathrm{pre}}(\hat{\mathbf{U}}\hat{\mathbf{U}}^\top g_{\mathrm{pre}}(\boldsymbol{x})) - h_{\mathrm{pre}}(\hat{\mathbf{U}}\hat{\mathbf{U}}^\top g_{\mathrm{exact}}(\boldsymbol{x}))\|_2 \\ &+ \|h_{\mathrm{pre}}(\hat{\mathbf{U}}\hat{\mathbf{U}}^\top g_{\mathrm{exact}}(\boldsymbol{x})) - h_{\mathrm{pre}}(\mathbf{0})\|_2 \\ &+ \|h_{\mathrm{pre}}(\mathbf{0}) - h_{\mathrm{exact}}(g_{\mathrm{exact}}(\boldsymbol{x}))\|_2, \end{aligned} \tag{11}$$

where $\mathbf{0}$ denotes a $d$-dimensional all-zero vector. We denote $B_1 = \|h_{\mathrm{pre}}(\hat{\mathbf{U}}\hat{\mathbf{U}}^\top g_{\mathrm{exact}}(\boldsymbol{x})) - h_{\mathrm{pre}}(\mathbf{0})\|_2$ and $C_1 = \|h_{\mathrm{pre}}(\mathbf{0}) - h_{\mathrm{exact}}(g_{\mathrm{exact}}(\boldsymbol{x}))\|_2$, which are given by

$$\begin{aligned} B_1 =& \|h_{\mathrm{pre}}(\hat{\mathbf{U}}\hat{\mathbf{U}}^\top g_{\mathrm{exact}}(\boldsymbol{x})) - h_{\mathrm{pre}}(\mathbf{0})\|_2 \\ \leq& L_{\mathrm{pre}} \cdot \|\hat{\mathbf{U}}\hat{\mathbf{U}}^\top g_{\mathrm{exact}}(\boldsymbol{x})\|_2 \quad \text{(Assumption 2)} \\ \leq& L_{\mathrm{pre}} \cdot (\|\mathbf{U}_*\mathbf{U}_*^\top g_{\mathrm{exact}}(\boldsymbol{x})\|_2 + \|(\hat{\mathbf{U}}\hat{\mathbf{U}}^\top - \mathbf{U}_*\mathbf{U}_*^\top)g_{\mathrm{exact}}(\boldsymbol{x})\|_2) \\ \leq& L_{\mathrm{pre}} \cdot (\epsilon_{\mathrm{fg}} + \mathcal{O}(\epsilon_{\mathrm{opt}})), \end{aligned} \tag{12}$$

where $\epsilon_{\mathrm{fg}}$ is from the Lemma 1, and $C_1 = \|h_{\mathrm{pre}}(\mathbf{0}) - h_{\mathrm{exact}}(g_{\mathrm{exact}}(\boldsymbol{x}))\|_2 \leq L_{\mathrm{exact}} \cdot \epsilon_{\mathrm{exact}}$ according to Assumption 1 and Assumption 2.

Therefore, given $\boldsymbol{x} \in \mathcal{D}$, the difference in outputs between $f_{\hat{\mathbf{U}}}$ and $f_{\mathrm{exact}}$ is bounded by $\|f_{\hat{\mathbf{U}}}(\boldsymbol{x}) - f_{\mathrm{exact}}(\boldsymbol{x})\|_2 \leq L_{\mathrm{pre}} \cdot (\epsilon_d + \max(\epsilon_{\mathrm{fg}}, \epsilon_{\mathrm{rm}}) + \mathcal{O}(\epsilon_{\mathrm{opt}})) + \epsilon_{\mathrm{exact}} \cdot \max(L_{\mathrm{exact}}, \epsilon_W)$. $\qquad\square$

## C  IMPLEMENTATION DETAILS OF EXPERIMENTS

### C.1  TRAINING DETAILS OF MULTI-CLASS UNLEARNING

For multi-class unlearning in Table 1 with Swin-T on Tiny-ImageNet, we adopt the PyTorch-released checkpoint trained on ImageNet-1K as a starting point, and fine-tune this checkpoint on Tiny-ImageNet to obtain the pretrained model. In the repeated 3 experiments of Table 1, the randomly selected 4 forgetting labels (classes) are $\{11, 83, 115, 153\}$, $\{44, 65, 150, 168\}$ and $\{53, 57, 108, 179\}$, respectively. All compared methods use the AdamW optimizer (Loshchilov & Hutter, 2019) with a weight decay of 0.05 and a batch size of 128. In L2UL (Cha et al., 2024), to generate adversarial examples, the $\ell_2$-PGD targeted attack is employed with a step size of 0.1, a perturbation bound of 0.4 and 100 iteration steps, and 200 adversarial examples are generated per image. The termination of the L2UL unlearning is carefully selected until the accuracy on the training forgetting data is sufficiently low without a substantial accuracy drop on the training remaining data. All the training hyper-parameters to obtain results in Table 1 are listed in Table A1. Our SUN adopts the Riemannian Adam optimizer (Kochurov et al., 2020) for the projection matrx $\mathbf{U}$ with a weight decay of 0.05, where the penultimate layer feature dimension with Swin-T on Tiny-ImageNet is $d = 768$.

### C.2  TRAINING DETAILS OF SINGLE-CLASS UNLEARNING

For single-class unlearning in Table 3 with ResNet50 on ImageNet-1K, we deploy the PyTorch-released checkpoint that is exactly pretrained on ImageNet-1K as the pretrained model $f_{\mathrm{pre}}$. Regarding the retrained model $f_{\mathrm{exact}}$ for unlearning, we train ResNet50 from scratch on the remaining data $\mathcal{D}_{\mathrm{rm}}$ on 4 NVIDIA GeForce RTX 4090 GPUs, which runs for around 24 hours. In the repeated 3 experiments of Table 3, the forgetting labels are 97, 316, and 852, respectively. All the compared methods use the SGD optimizer (Bottou, 2012) with a weight decay of $1 \times 10^{-4}$, the momentum of 0.9, and a batch size of 128. All the training hyper-parameters for the results in Table 3 are listed Table A2. Our SUN keeps the settings of using the Riemannian Adam optimizer (Kochurov et al., 2020) for $\mathbf{U}$ with a weight decay of 0.05, where the penultimate layer feature dimension with ResNet50 on ImageNet-1K is $d = 2048$.

Table A1: Training hyper-parameters of different MU methods w.r.t. results of multi-class unlearning in Table 1 with Swin-T on Tiny-ImageNet.

| method | $\mathcal{D}_{\mathrm{rm}}$ | $\mathcal{D}_{\mathrm{fg}}$ | hyper-parameters |
|---|---|---|---|
| pretrained | ✓ | ✓ | 20 epochs, lr $= 10^{-4}$, cosine scheduler |
| retrained | ✓ | ✗ | 20 epochs, lr $= 10^{-4}$, cosine scheduler |
| FT | ✓ | ✗ | 10 epochs, lr $= 10^{-4}$, cosine scheduler |
| GA | ✗ | ✓ | 10 epochs, lr $= 2 \times 10^{-6}$, constant scheduler |
| RL | ✗ | ✓ | 10 epochs, lr $= 10^{-5}$, cosine scheduler |
| RL | ✓ | ✓ | 10 epochs, lr $= 10^{-4}$, cosine scheduler |
| SalUn | ✗ | ✓ | 10 epochs, lr $= 10^{-5}$, cosine scheduler, saliency sparsity 50% |
| SalUn | ✓ | ✓ | 10 epochs, lr $= 10^{-4}$, cosine scheduler, saliency sparsity 50% |
| BT | ✗ | ✓ | 10 epochs, lr $= 10^{-5}$, cosine scheduler, temperature scalar = 1.0 |
| L2UL | ✗ | ✓ | lr $= 10^{-5}$, constant scheduler, regularization coefficient = 1.0 |
| DELETE | ✗ | ✓ | 10 epochs, lr $= 10^{-5}$, cosine scheduler |
| **SUN** | only $\boldsymbol{\Sigma}_{\mathrm{rm}}^{\mathrm{pre}}$ | only $\boldsymbol{\Sigma}_{\mathrm{fg}}^{\mathrm{pre}}$ | 50 steps, $s = 250$, lr $= 1$, constant scheduler |

Table A2: Training hyper-parameters of different MU methods w.r.t. results of single-class unlearning in Table 3 with ResNet50 on ImageNet-1K.

| method | $\mathcal{D}_{\mathrm{rm}}$ | $\mathcal{D}_{\mathrm{fg}}$ | hyper-parameters |
|---|---|---|---|
| pretrained | ✓ | ✓ | - |
| retrained | ✓ | ✗ | 90 epochs, lr $= 10^{-1}$, LR decay: 0.1 every 30 epochs |
| GA | ✗ | ✓ | 3 epochs, lr $= 10^{-4}$, constant scheduler |
| RL | ✗ | ✓ | 5 epochs, lr $= 5 \times 10^{-5}$, cosine scheduler |
| SalUn | ✗ | ✓ | 10 epochs, lr $= 10^{-4}$, cosine scheduler, saliency sparsity 50% |
| BT | ✗ | ✓ | 5 epochs, lr $= 1 \times 10^{-5}$, cosine scheduler, temperature scalar = 1.0 |
| DELETE | ✗ | ✓ | 5 epochs, lr $= 10^{-1}$, cosine scheduler |
| **SUN** | only $\boldsymbol{\Sigma}_{\mathrm{rm}}^{\mathrm{pre}}$ | only $\boldsymbol{\Sigma}_{\mathrm{fg}}^{\mathrm{pre}}$ | 100 steps, $s = 500$, lr $= 10$, constant scheduler |

## C.3 TRAINING DETAILS OF EXTREME UNLEARNING

For results of extreme unlearning in Table 4 with Swin-T on Tiny-ImageNet, the basic settings are generally similar with that in Sec. C.1. In the repeated 3 experiments of Table 4, we adopt different random seeds, i.e., 0, 1 and 2, to guarantee that the forgetting 180 labels are different in each experiment. All compared methods use the AdamW optimizer (Loshchilov & Hutter, 2019) with weight decay of 0.05 and a batch size of 128. The hyper-parameters for the results in Table 4 are listed in Table A3. In Table A3, the $\mathcal{D}_{\mathrm{fg}}$-based GA and SalUn involving both $\mathcal{D}_{\mathrm{rm}}$ and $\mathcal{D}_{\mathrm{fg}}$ are highly-sensitive to the choice of the learning rate under this extreme unlearning scenario. To obtain satisfactory results, we use different learning rates for GA and SalUn in each experiment w.r.t. different random seeds. For GA, the hyper-parameters are $\{4$ epochs, lr $= 3 \times 10^{-7}\}$, $\{7$ epochs, lr $= 2 \times 10^{-7}\}$ and $\{3$ epochs, lr $= 4 \times 10^{-7}\}$ w.r.t. random seeds 0, 1 and 2. For SalUn based on $\mathcal{D}_{\mathrm{rm}}$ and $\mathcal{D}_{\mathrm{fg}}$, the hyper-parameters are $\{$lr $= 2 \times 10^{-4}\}$, $\{$lr $= 1.5 \times 10^{-4}\}$ and $\{$lr $= 10^{-4}\}$ w.r.t. random seeds 0, 1 and 2. For L2UL, we only generate two adversarial examples per image for efficiency. Our SUN optimizes $\mathbf{U}$ through the Riemannian Adam optimizer (Kochurov et al., 2020) with the weight decay of 0.05.

## C.4 TRAINING DETAILS OF INSTANCE UNLEARNING

For instance unlearning in Table 5 with Swin-T on Tiny-ImageNet, the basic settings are with that in Sec. C.1. 1000 from the total 100,000 training samples are randomly selected as the forgetting data $\mathcal{D}_{\mathrm{fg}}$ (1%). Each of the repeated 3 experiments of Table 5 is corresponding to different random seeds, i.e., 0, 1 and 2. All compared methods use the AdamW optimizer (Loshchilov & Hutter, 2019) with a weight decay of 0.05 and a batch size of 128. The hyper-parameters are listed in Table A4. Our SUN uses the Riemannian Adam optimizer (Kochurov et al., 2020) with a weight decay of 0.05.

## D ADDITIONAL EXPERIMENTAL RESULTS

### D.1 CONTINUAL UNLEARNING

**Settings.** We provide evaluations for different MU methods under a practical scenario where multiple unlearning requests are in a continual order. To be specific, we consider a 4-round continual

Table A3: Training hyper-parameters of different MU methods w.r.t. results of extreme unlearning in Table 4 with Swin-T on Tiny-ImageNet.

| method | $\mathcal{D}_{\mathrm{rm}}$ | $\mathcal{D}_{\mathrm{fg}}$ | hyper-parameters |
|---|---|---|---|
| pretrained | ✓ | ✓ | 20 epochs, lr $= 10^{-4}$, cosine scheduler |
| retrained | ✓ | ✗ | 20 epochs, lr $= 10^{-4}$, cosine scheduler |
| FT | ✓ | ✗ | 10 epochs, lr $= 10^{-4}$, cosine scheduler |
| GA | ✗ | ✓ | constant scheduler |
| RL | ✗ | ✓ | 5 epochs, lr $= 5 \times 10^{-7}$, cosine scheduler |
| RL | ✓ | ✓ | 10 epochs, lr $= 2 \times 10^{-5}$, cosine scheduler |
| SalUn | ✗ | ✓ | 5 epochs, lr $= 5 \times 10^{-7}$, cosine scheduler, saliency sparsity 50% |
| SalUn | ✓ | ✓ | 10 epochs, cosine scheduler, saliency sparsity 50% |
| BT | ✗ | ✓ | 2 epochs, lr $= 10^{-6}$, cosine scheduler, temperature scalar = 1.0 |
| L2UL | ✗ | ✓ | lr $= 2 \times 10^{-4}$, constant scheduler, regularization coefficient = 1.0 |
| DELETE | ✗ | ✓ | 10 epochs, lr $= 2 \times 10^{-4}$, cosine scheduler |
| **SUN** | only $\mathbf{\Sigma}_{\mathrm{rm}}^{\mathrm{pre}}$ | only $\mathbf{\Sigma}_{\mathrm{fg}}^{\mathrm{pre}}$ | 50 steps, $s = 18$, lr $= 1$, constant scheduler |

Table A4: Training hyper-parameters of different MU methods w.r.t. results of instance unlearning in Table 5 with Swin-T on Tiny-ImageNet.

| method | $\mathcal{D}_{\mathrm{rm}}$ | $\mathcal{D}_{\mathrm{fg}}$ | hyper-parameters |
|---|---|---|---|
| pretrained | ✓ | ✓ | 20 epochs, lr $= 10^{-4}$, cosine scheduler |
| retrained | ✓ | ✗ | 20 epochs, lr $= 10^{-4}$, cosine scheduler |
| GA | ✗ | ✓ | 10 epochs, lr $= 6 \times 10^{-6}$ constant scheduler |
| RL | ✗ | ✓ | 10 epochs, lr $= 10^{-5}$, cosine scheduler |
| SalUn | ✗ | ✓ | 10 epochs, lr $= 10^{-5}$, cosine scheduler, saliency sparsity 50% |
| BT | ✗ | ✓ | 10 epochs, lr $= 10^{-5}$, cosine scheduler, temperature scalar = 1.0 |
| L2UL | ✗ | ✓ | lr $= 10^{-4}$, constant scheduler, regularization coefficient = 1.0 |
| DELETE | ✗ | ✓ | 10 epochs, lr $= 2 \times 10^{-5}$, cosine scheduler |
| **SUN** | only $\mathbf{\Sigma}_{\mathrm{rm}}^{\mathrm{pre}}$ | only $\mathbf{\Sigma}_{\mathrm{fg}}^{\mathrm{pre}}$ | 100 steps, $s = 300$, lr $= 1$, constant scheduler |

unlearning. In each round, 2 classes (1%) are randomly selected as the forgetting data $\mathcal{D}_{\mathrm{fg}}$, and the forgetting labels at each round are $\{18,170\}$, $\{80,49\}$, $\{117,51\}$, and $\{16,118\}$, respectively. We introduce two new metrics $\mathrm{Acc}_{\mathrm{fgp}}^{\mathrm{tr}}$ and $\mathrm{Acc}_{\mathrm{fgp}}^{\mathrm{te}}$, i.e., accuracy on the forgetting data in previous rounds $\mathcal{D}_{\mathrm{fgp}}$, to evaluate whether the model indeed forget data from early unlearning requests. Results of continual unlearning are shown in Table A5 including multiple mainstream MU methods.

In this continual unlearning, our SUN optimizes to guarantee the unlearning performance on forgetting data from previous rounds $\mathcal{D}_{\mathrm{fgp}}$, where we introduce an the loss $J_{\mathrm{fgp}}$ to the objective of $J(\mathbf{U})$ in (2) for achieving the task of continual unlearning as follows:

$$\min_{\mathbf{U} \in \mathrm{St}(d,s)} \underbrace{\left(\frac{\mathrm{Tr}(\mathbf{U}^\top \mathbf{\Sigma}_{\mathrm{fgp}}^{\mathrm{pre}} \mathbf{U})}{\mathrm{Tr}(\mathbf{\Sigma}_{\mathrm{fgp}}^{\mathrm{pre}})}\right)^2}_{J_{\mathrm{fgp}}} + \underbrace{\left(\frac{\mathrm{Tr}(\mathbf{U}^\top \mathbf{\Sigma}_{\mathrm{fg}}^{\mathrm{pre}} \mathbf{U})}{\mathrm{Tr}(\mathbf{\Sigma}_{\mathrm{fg}}^{\mathrm{pre}})}\right)^2}_{J_{\mathrm{fg}}} + \underbrace{\left(\frac{\mathrm{Tr}\left(\mathbf{\Sigma}_{\mathrm{rm}}^{\mathrm{pre}} - \mathbf{U}\mathbf{U}^\top \mathbf{\Sigma}_{\mathrm{rm}}^{\mathrm{pre}} \mathbf{U}\mathbf{U}^\top\right)}{\mathrm{Tr}\left(\mathbf{\Sigma}_{\mathrm{rm}}^{\mathrm{pre}}\right)}\right)^2}_{J_{\mathrm{rm}}}. \quad (13)$$

$J_{\mathrm{fgp}}$ takes a similar form as $J_{\mathrm{fg}}$ and measures the projected variance of the covariance matrix $\mathbf{\Sigma}_{\mathrm{fgp}}^{\mathrm{pre}}$ w.r.t. $\mathcal{D}_{\mathrm{fgp}}$ captured within the subspace spanned by $\mathbf{U}$. Hence, minimizing $J_{\mathrm{fgp}}$ achieves unlearning on $\mathcal{D}_{\mathrm{fgp}}$. Note that $J_{\mathrm{fgp}}$ can be efficiently implemented without additional computation, since $\mathbf{\Sigma}_{\mathrm{fgp}}^{\mathrm{pre}}$ has been accessed in previous rounds.

**Results.** During the multiple rounds of unlearning, other MU methods constantly modify the parameters of the pretrained model $f_{\mathrm{pre}}$ and directly affect the learning ability of $f_{\mathrm{pre}}$. As a result, in Table A5, these methods show significant accuracy drops on the remaining data $\mathcal{D}_{\mathrm{rm}}$ and even slight accuracy increases on the forgetting data $\mathcal{D}_{\mathrm{fgp}}$ in previous rounds, after 4 rounds of unlearning. In contrast, our SUN does not update the backbone $g_{\mathrm{pre}}$ and the linear layer $h_{\mathrm{pre}}$ of the pretrained $f_{\mathrm{pre}}$, and well preserves the learning ability of $f_{\mathrm{pre}}$. By only optimizing the projection matrix $\mathbf{U}$ given $\mathbf{\Sigma}_{\mathrm{rm}}^{\mathrm{pre}}$, $\mathbf{\Sigma}_{\mathrm{fg}}^{\mathrm{pre}}$ and $\mathbf{\Sigma}_{\mathrm{fgp}}^{\mathrm{pre}}$, the learned subspace of SUN successfully maintains the accuracy on $\mathcal{D}_{\mathrm{rm}}$ and reduces the accuracy on $\mathcal{D}_{\mathrm{fg}}$ and $\mathcal{D}_{\mathrm{fgp}}$.

Table A5: Comparison results of Swin-T on Tiny-ImageNet under continual unlearning.

| method | $\mathcal{D}_{\mathrm{rm}}$ | $\mathcal{D}_{\mathrm{fg}}$ | $\mathcal{D}_{\mathrm{fgp}}$ | $\mathrm{Acc}_{\mathrm{rm}}^{\mathrm{tr}}$ | $\mathrm{Acc}_{\mathrm{fg}}^{\mathrm{tr}}$ | $\mathrm{Acc}_{\mathrm{fgp}}^{\mathrm{tr}}$ | $\mathrm{Acc}_{\mathrm{rm}}^{\mathrm{te}}$ | $\mathrm{Acc}_{\mathrm{fg}}^{\mathrm{te}}$ | $\mathrm{Acc}_{\mathrm{fgp}}^{\mathrm{te}}$ |
|---|---|---|---|---|---|---|---|---|---|
| | | | | *Round-1* | | | | | |
| pretrained | ✓ | ✓ | - | 99.63 | 99.90 | - | 74.49 | 80.00 | - |
| retrained | ✓ | ✗ | - | 99.65 (0.00) | 0.00 (0.00) | - | 75.08 (0.00) | 0.00 (0.00) | - |
| RL | ✗ | ✓ | - | 97.00 (2.65) | 8.50 (8.50) | - | 70.87 (4.21) | 1.00 (1.00) | - |
| RL | ✓ | ✓ | - | 99.93 (0.28) | 4.90 (4.90) | - | 74.34 (0.74) | 2.00 (2.00) | - |
| BT | ✗ | ✓ | - | 98.57 (1.08) | 21.60 (21.60) | - | 72.58 (2.50) | 6.00 (6.00) | - |
| DELETE | ✗ | ✓ | - | 99.21 (0.44) | 8.00 (8.00) | - | 73.87 (0.62) | 0.00 (0.00) | - |
| **SUN** | $\Sigma_{\mathrm{rm}}^{\mathrm{pre}}$ | $\Sigma_{\mathrm{fg}}^{\mathrm{pre}}$ | - | 96.86 (2.77) | 4.20 (4.20) | - | 72.65 (2.43) | 0.00 (0.00) | - |
| | | | | *Round-2* | | | | | |
| pretrained | ✓ | ✓ | ✓ | 99.63 | 99.60 | 99.90 | 74.55 | 69.00 | 80.00 |
| retrained | ✓ | ✗ | ✗ | 99.69 (0.00) | 0.00 (0.00) | 0.00 (0.00) | 75.57 (0.00) | 0.00 (0.00) | 0.00 (0.00) |
| RL | ✗ | ✓ | ✗ | 80.30 (19.39) | 6.30 (6.30) | 1.00 (1.00) | 59.68 (15.89) | 2.00 (2.00) | 0.00 (0.00) |
| RL | ✓ | ✓ | ✗ | 99.98 (0.29) | 6.30 (6.30) | 1.00 (1.00) | 74.37 (1.20) | 1.00 (1.00) | 0.00 (0.00) |
| BT | ✗ | ✓ | ✗ | 91.42 (8.21) | 20.30 (20.30) | 6.60 (6.60) | 67.07 (8.50) | 8.00 (8.00) | 3.00 (3.00) |
| DELETE | ✗ | ✓ | ✗ | 98.33 (1.30) | 10.60 (10.60) | 3.90 (3.90) | 72.40 (3.17) | 5.00 (5.00) | 0.00 (0.00) |
| **SUN** | $\Sigma_{\mathrm{rm}}^{\mathrm{pre}}$ | $\Sigma_{\mathrm{fg}}^{\mathrm{pre}}$ | $\Sigma_{\mathrm{fgp}}^{\mathrm{pre}}$ | 95.77 (3.92) | 7.50 (7.50) | 0.50 (0.50) | 72.01 (3.56) | 4.00 (4.00) | 1.00 (1.00) |
| | | | | *Round-3* | | | | | |
| pretrained | ✓ | ✓ | ✓ | 99.62 | 99.60 | 99.75 | 74.53 | 69.00 | 74.50 |
| retrained | ✓ | ✗ | ✗ | 99.66 (0.00) | 0.00 (0.00) | 0.00 (0.00) | 73.06 (0.00) | 0.00 (0.00) | 0.00 (0.00) |
| RL | ✗ | ✓ | ✗ | 45.16 (54.50) | 0.70 (0.70) | 0.30 (0.30) | 36.27 (36.79) | 0.00 (0.00) | 0.50 (0.50) |
| RL | ✓ | ✓ | ✗ | 99.98 (0.32) | 7.70 (7.70) | 0.30 (0.30) | 73.79 (0.73) | 2.00 (2.00) | 0.00 (0.00) |
| BT | ✗ | ✓ | ✗ | 72.06 (27.60) | 6.40 (6.40) | 4.45 (4.45) | 54.42 (18.64) | 1.00 (1.00) | 2.50 (2.50) |
| DELETE | ✗ | ✓ | ✗ | 96.27 (3.39) | 5.10 (5.10) | 3.85 (3.85) | 70.05 (3.01) | 3.00 (3.00) | 1.50 (1.50) |
| **SUN** | $\Sigma_{\mathrm{rm}}^{\mathrm{pre}}$ | $\Sigma_{\mathrm{fg}}^{\mathrm{pre}}$ | $\Sigma_{\mathrm{fgp}}^{\mathrm{pre}}$ | 96.39 (3.27) | 1.70 (1.70) | 0.35 (0.35) | 72.80 (0.26) | 0.00 (0.00) | 0.50 (0.50) |
| | | | | *Round-4* | | | | | |
| pretrained | ✓ | ✓ | ✓ | 99.63 | 99.30 | 99.80 | 74.54 | 73.00 | 75.33 |
| retrained | ✓ | ✗ | ✗ | 99.67 (0.00) | 0.00 (0.00) | 0.00 (0.00) | 75.72 (0.00) | 0.00 (0.00) | 0.00 (0.00) |
| RL | ✗ | ✓ | ✗ | 10.96 (88.71) | 0.30 (0.30) | 0.00 (0.00) | 9.97 (65.75) | 1.00 (1.00) | 0.00 (0.00) |
| RL | ✓ | ✓ | ✗ | 99.98 (0.31) | 17.60 (17.60) | 1.07 (1.07) | 73.43 (2.29) | 9.00 (9.00) | 0.00 (0.00) |
| BT | ✗ | ✓ | ✗ | 47.28 (52.39) | 3.00 (3.00) | 1.63 (1.63) | 37.69 (38.03) | 1.00 (1.00) | 0.67 (0.67) |
| DELETE | ✗ | ✓ | ✗ | 93.07 (6.60) | 3.80 (3.80) | 2.43 (2.43) | 67.91 (7.81) | 3.00 (3.00) | 1.33 (1.33) |
| **SUN** | $\Sigma_{\mathrm{rm}}^{\mathrm{pre}}$ | $\Sigma_{\mathrm{fg}}^{\mathrm{pre}}$ | $\Sigma_{\mathrm{fgp}}^{\mathrm{pre}}$ | 96.48 (3.19) | 5.50 (5.50) | 0.07 (0.07) | 72.50 (3.22) | 3.00 (3.00) | 0.00 (0.00) |

Table A6: Hyper-parameters in different positions for subspace learning.

| position | $d$ | $s$ | steps |
|---|---|---|---|
| layer1 | 4,096 | 500 | 500 |
| layer2 | 2,048 | 350 | 200 |
| layer3 | 1,024 | 9 | 200 |
| layer4 | 512 | 9 | 50 |

# E  IMPLEMENTATION DETAILS AND ADDITIONAL RESULTS OF APPLICATIONS

## E.1  IMPLEMENTATION POSITIONS FOR SUBSPACE LEARNING

Our SUN can be flexibly implemented in different positions of the pretrained model. Table A6 provides the corresponding feature dimensions and the optimization steps when applying SUN in different layer modules, which is in relation to the sensitivity analysis results of the right panel in Fig. 3 with ResNet18 on CIFAR10.

## E.2  FACE RECOGNITION WITH MACHINE UNLEARNING

**Settings.**  Experiments are conducted on a prevalent face identity recognition dataset VGGFace2 (Cao et al., 2018), including approximately 3.31 million images collected from 9,131 identities. Particularly, we adopt its publicly-available version[2], where all face images are aligned and cropped to $112 \times 112 \times 3$. Our experiments utilize a filtered set of 200 identities, each with over 500 images. Then, we randomly select 400 images for each identity, forming a training set containing 80,000 images, and the rest is adopted as the test set. One of the 200 identities is randomly selected as the forgetting data $\mathcal{D}_{\mathrm{fg}}$, which is a challenging setting for class-centric unlearning. The adopted neural network model is ResNet50 (He et al., 2016). All the compared methods use the SGD optimizer (Bottou, 2012) with a weight decay of $5 \times 10^{-4}$, the momentum of 0.9, and a batch size of 128. More training details are in Table A7 with comparison results in Table 6 in the main context.

**Results.**  This unlearning setting, i.e., forgetting one of the 200 identities, implies a substantial size of $\mathcal{D}_{\mathrm{rm}}$. Therefore, MU methods involving $\mathcal{D}_{\mathrm{rm}}$ can easily achieve the ideal performance of the exact unlearning model of $f_{\mathrm{exact}}$, as illustrated by the FT, RL, and SalUn in Table 6. In contrast,

---

[2]https://www.kaggle.com/datasets/yakhyokhuja/vggface2-112x112

Table A7: Training hyper-parameters of different MU methods w.r.t. Table 6 with ResNet50 on VGGFace2.

| method | $\mathcal{D}_{\mathrm{rm}}$ | $\mathcal{D}_{\mathrm{fg}}$ | hyper-parameters |
|---|---|---|---|
| pretrained | ✓ | ✓ | 200 epochs, lr $= 10^{-1}$, cosine scheduler |
| retrained | ✓ | ✗ | 200 epochs, lr $= 10^{-1}$, cosine scheduler |
| FT | ✓ | ✗ | 10 epochs, lr $= 5 \times 10^{-2}$, cosine scheduler |
| GA | ✗ | ✓ | 5 epochs, lr $= 5 \times 10^{-4}$, constant scheduler |
| RL | ✗ | ✓ | 10 epochs, lr $= 5 \times 10^{-4}$, cosine scheduler |
| RL | ✓ | ✓ | 10 epochs, lr $= 5 \times 10^{-2}$, cosine scheduler |
| SalUn | ✗ | ✓ | 10 epochs, lr $= 10^{-4}$, cosine scheduler, saliency sparsity 50% |
| SalUn | ✓ | ✓ | 10 epochs, lr $= 5 \times 10^{-2}$, cosine scheduler, saliency sparsity 50% |
| BT | ✗ | ✓ | 15 epochs, lr $= 5 \times 10^{-4}$, cosine scheduler, temperature scalar = 1.0 |
| L2UL | ✗ | ✓ | lr $= 1 \times 10^{-4}$, constant scheduler, regularization coefficient = 1.0 |
| DELETE | ✗ | ✓ | 10 epochs, lr $= 5 \times 10^{-4}$, cosine scheduler |
| **SUN** | only $\boldsymbol{\Sigma}_{\mathrm{rm}}^{\mathrm{pre}}$ | only $\boldsymbol{\Sigma}_{\mathrm{fg}}^{\mathrm{pre}}$ | 50 steps, $s = 21$, lr $= 1$, constant scheduler |

Table A8: Training hyper-parameters of different MU methods w.r.t. Table A9 with ResNet18 on RAF-DB.

| method | $\mathcal{D}_{\mathrm{rm}}$ | $\mathcal{D}_{\mathrm{fg}}$ | hyper-parameters |
|---|---|---|---|
| pretrained | ✓ | ✓ | 200 epochs, lr $= 10^{-1}$, cosine scheduler |
| retrained | ✓ | ✗ | 200 epochs, lr $= 10^{-1}$, cosine scheduler |
| FT | ✓ | ✗ | 10 epochs, lr $= 5 \times 10^{-2}$, cosine scheduler |
| GA | ✗ | ✓ | 10 epochs, lr $= 5 \times 10^{-4}$, constant scheduler |
| RL | ✗ | ✓ | 20 epochs, lr $= 5 \times 10^{-4}$, cosine scheduler |
| RL | ✓ | ✓ | 10 epochs, lr $= 5 \times 10^{-2}$, cosine scheduler |
| SalUn | ✗ | ✓ | 20 epochs, lr $= 10^{-4}$, cosine scheduler, saliency sparsity 50% |
| SalUn | ✓ | ✓ | 10 epochs, lr $= 5 \times 10^{-2}$, cosine scheduler, saliency sparsity 50% |
| BT | ✗ | ✓ | 15 epochs, lr $= 5 \times 10^{-4}$, cosine scheduler, temperature scalar = 1.0 |
| L2UL | ✗ | ✓ | lr $= 2 \times 10^{-4}$, constant scheduler, regularization coefficient = 1.0 |
| DELETE | ✗ | ✓ | 20 epochs, lr $= 2 \times 10^{-4}$, cosine scheduler |
| **SUN** | only $\boldsymbol{\Sigma}_{\mathrm{rm}}^{\mathrm{pre}}$ | only $\boldsymbol{\Sigma}_{\mathrm{fg}}^{\mathrm{pre}}$ | 50 steps, $s = 6$, lr $= 1$, constant scheduler |

it is challenging for MU methods by only accessing $\mathcal{D}_{\mathrm{fg}}$ under this unlearning setting, as demonstrated by the relatively high accuracy on $\mathcal{D}_{\mathrm{fg}}$ of RL, SalUN, BT, L2UL and DELETE in Table 6. Our SUN does not visit the original samples and learns to preserve and remove essential patterns from the covariance matrices of $\mathcal{D}_{\mathrm{rm}}$ and $\mathcal{D}_{\mathrm{fg}}$ through low-dimensional projections, managing to simultaneously maintain high accuracy on $\mathcal{D}_{\mathrm{rm}}$ and suppress accuracy on $\mathcal{D}_{\mathrm{fg}}$.

### E.3 EMOTION RECOGNITION WITH MACHINE UNLEARNING

**Settings.** Experiments are conducted on a popular emotion recognition dataset RAF-DB (Li et al., 2017), RAF-DB contains 12,271 and 3,068 training and test images with a size of $100 \times 100 \times 3$, respectively, within 7 different emotions: "surprise", "fear", "disgust", "happiness", "sadness", "anger" and "neutral". In the experiments, one emotion is randomly selected as the forgetting data $\mathcal{D}_{\mathrm{fg}}$ ("fear"). The adopted model is ResNet18 (He et al., 2016). All the compared methods use the SGD optimizer (Bottou, 2012) with a weight decay of $5 \times 10^{-4}$, the momentum of 0.9, and a batch size of 128. More training details are listed Table A8. The comparison results among different MU methods are shown in Table A9.

**Results.** As illustrated in Table A9, when forgetting the "fear" emotion from the total 7 emotions, SUN shows the best approximation performance towards that of the retrained model $f_{\mathrm{exact}}$. Other MU methods either heavily rely on the remaining data $\mathcal{D}_{\mathrm{rm}}$ (RL and SalUn), or show unsatisfactory results on $\mathcal{D}_{\mathrm{fg}}$ (BT, L2UL and DELETE). SUN explores the low-dimensional subspace in which the features of $\mathcal{D}_{\mathrm{fg}}$ and $\mathcal{D}_{\mathrm{rm}}$ can be well distinguished through its reconstruction performances, achieving distinctively better unlearning performances.

## F USE OF LARGE LANGUAGE MODELS

The use of large language models in this work was minimal and restricted solely to textual polishing in the preparation of this submission.

Table A9: Comparison results of ResNet18 on RAF-DB.

| method | $\mathcal{D}_{\mathrm{rm}}$ | $\mathcal{D}_{\mathrm{fg}}$ | $\mathbf{Acc_{rm}^{tr}}$ | $\mathbf{Acc_{fg}^{tr}}$ | $\mathbf{Acc_{rm}^{te}}$ | $\mathbf{Acc_{fg}^{te}}$ | **MIA** | Avg.G. |
|---|---|---|---|---|---|---|---|---|
| pretrained | ✓ | ✓ | 100.00 | 100.00 | 85.10 | 56.76 | 100.00 | - |
| retrained | ✓ | ✗ | 100.00 (0.00) | 0.00 (0.00) | 84.90 (0.00) | 0.00 (0.00) | 0.00 (0.00) | 0.00 |
| GA | ✗ | ✓ | 91.38 (8.62) | 0.36 (0.36) | 76.75 (8.15) | 0.00 (0.00) | 0.36 (0.36) | 3.50 |
| FT | ✓ | ✗ | 99.97 (0.03) | 0.00 (0.00) | 83.97 (0.03) | 0.00 (0.00) | 0.00 (0.00) | 0.01 |
| RL | ✗ | ✓ | 98.96 (1.04) | 15.66 (15.66) | 79.93 (4.97) | 5.41 (5.41) | 0.00 (0.00) | 5.42 |
| RL | ✓ | ✓ | 99.92 (0.08) | 0.00 (0.00) | 84.10 (0.80) | 0.00 (0.00) | 0.00 (0.00) | 0.18 |
| SalUn | ✗ | ✓ | 99.83 (0.17) | 12.46 (12.46) | 82.53 (2.37) | 2.70 (2.70) | 0.36 (0.36) | 3.61 |
| SalUn | ✓ | ✓ | 99.98 (0.02) | 0.00 (0.00) | 84.57 (0.33) | 0.00 (0.00) | 0.00 (0.00) | 0.07 |
| BT | ✗ | ✓ | 99.83 (0.17) | 10.68 (10.68) | 82.63 (2.27) | 2.70 (2.70) | 1.07 (1.07) | 3.38 |
| L2UL | ✗ | ✓ | 99.99 (0.01) | 10.68 (10.68) | 84.03 (0.87) | 1.35 (1.35) | 0.00 (0.00) | 2.58 |
| DELETE | ✗ | ✓ | 99.72 (0.28) | 9.25 (9.25) | 82.20 (2.70) | 2.70 (2.70) | 1.42 (1.42) | 3.27 |
| **SUN** | only $\mathbf{\Sigma_{rm}^{pre}}$ | only $\mathbf{\Sigma_{fg}^{pre}}$ | 100.00 (**0.00**) | 0.00 (**0.00**) | 84.90 (**0.00**) | 0.00 (**0.00**) | 0.00 (**0.00**) | **0.00** |

