# OpenReview forum: "Machine Unlearning in Low-dimensional Feature Subspace"
_ICLR.cc/2026/Conference — Submitted to ICLR 2026_

### Official Review · Reviewer_2Z6X · 2025-10-21

**Soundness:** 2
**Presentation:** 3
**Contribution:** 1
**Rating:** 4
**Confidence:** 4

**Summary:**

This paper proposes a feature-level plug-in module to achieve machine unlearning. The core idea is based on the observation that features from different data sources exhibit stronger separability at the feature level. Leveraging this property, the method projects features into a subspace where the representations of the data to be forgotten are minimized as much as possible. Finally, the features are projected back into the original space to enable subsequent inference.

**Strengths:**

1. The paper exhibits a well-structured organization. Particularly in sections introducing the methodology, the presentation logically unfolds the implementation process of the proposed method.

2. The experimental part provides well-structured comparisons. Through these comparisons, the differences and enhancement effects of the proposed SUN method in critical indicators.

**Weaknesses:**

1. The design of such a plug-in seems to contradict the machine unlearning, and I still remain convinced about this approach. The target of unlearning is to resolve the impact of the forgetting data. However, plug-in only introduces a new modules and the vanilla weights are still saved in the model. If attackers continue to concatenate these to form a new model, will it not still be capable of recovering the ability to acquire the forgotten knowledge? It appears that this solution cannot be deemed effective when viewed from this perspective.

2. The author claims that the motivation comes from the idea that different data points are more separable in high-dimensional feature spaces. However, this claim does not seem to be theoretically proven or empirically validated in the paper. Moreover, this motivation is not necessarily intuitively correct. For classification tasks, where data are separated based on class labels, this reasoning might hold to some extent. However, for broader types of tasks, the motivation may not be valid. The discussion of this point in the paper is not sufficiently in-depth.

3. Even assuming that the motivation is partially correct, how should one decide which features to use and from which layer? The paper lacks sufficient ablation studies on the application of this module, as well as relevant comparative analyses. For example, how do features from different layers differ? How should the corresponding configurations be chosen under different model architectures?

4. The experimental performance does not appear to be sota. In some experiments, the results are inferior to those of existing methods.

5. The PCA projection method is overly simplistic. If the dataset is large in scale, PCA may not be feasible in terms of dimensionality. For example, when separating 10w data points in a 1k-dimensional subspace, substantial overlap is likely to occur.

**Questions:**

1. Could the authors provide some theoretical justification and empirical validation for this motivation? At present, this motivation does not seem self-evident.

2. Could the authors discuss the separation efficiency of the features after PCA? For instance, they could use cosine similarity or other metrics to illustrate the relationship among dimensionality, dataset size, and separation efficiency.

3. I noticed that the experiments in the paper mainly focus on classification baselines. This is related to the issues I mentioned earlier — the separability of the feature space is inherently tied to the category-based nature of classification tasks. However, for tasks in other domains, such as NLP, this approach may not be effective, since the feature spaces in those cases may not exhibit strong separability. Could the authors provide some relevant experiments to support the generality of their method?

4. Could the authors evaluate the performance of their method on some smaller networks, such as encoder–decoder or VAE architectures? The effectiveness of PCA becomes limited when dealing with a large number of categories or high-dimensional vectors. In current large-scale language models, the vocabulary size often reaches 100K or even 1M, and the vector dimensionality can exceed 10K. How does this method affect the transferability and performance of such large-scale models?

---

> ### Author Response · Authors · 2025-11-21
> **Response 7**
>
> Dear Reviewer 2Z6X,
>
> We appreciate your efforts and constructive feedback that help improve our work.
>
> In the following, we make pointwise responses to the raised comments, which we hope could resolve your concerns. We note **[W]** for weaknesses and **[Q]** for questions.
>
>
> We would look forward to fruitful discussions with you! Thanks!
>
> Kind regards,
> The Authors
>
> ---
> **W1**: *Attackers being aware of the plug-in module and the model weights.*
>
> > We thank the reviewer for this interesting discussion.
> >
> > The plug-in module operates on the penultimate-layer features to mitigate the influence of the forgetting data and meanwhile maintains the information of remaining data, i.e., the dual targets in machine unlearning, such that  final model outputs contain little knowledge from the forgetting data.  From the view of  dual targets, it does not contradict with machine unlearning, though the "*forgetting*" in our method is not reflected by the change of all model parameters.
> >
> >
> > We agree that when attackers jointly have access to *(1)* the backbone, *(2)* our plug-in module, and *(3)* the  position of our plug-in module which can be implemented in different layers as evaluated *in the right panel in Figure 3*, it is possible to recover the forgetting knowledge from this perspective. Nevertheless, *the case that the attacker is aware of all information of (1), (2) and (3) is rare in practice*, as this is an extremely risky scenario of privacy leakage. In principle, any unlearning method would be vulnerable to privacy leakage if a white-box attacker had full knowledge of its internal unlearning mechanism.
> >
> > Besides, we could informally consider another possibility for privacy leakage during the unlearning stage: existing methods iteratively visit and compute the parameter gradients of the whole network, hence the attackers could have more chances to access such process to get/estimate gradients, posing privacy leakage during unlearning. Our method avoids such process during unlearning and thus has less exposure to model information, which could be to some extent a way to mitigate privacy leakage in this sense. This is though not yet the main focus of this work  but worth further works addressing  such scenarios.
>
> **W2 & Q1**: *The motivation is not theoretically proven or empirically validated.*
>
> > **(1) Empirically**, *reconstruction error analyses* are conducted to justify our motivation for feature subspace in **Line 200** - **Line 208** and **Figure 2**, and *ablation studies* on the established subspace optimization are presented in **Line 460** - **Line 470** and **Table 7** for relevant justification.
> >
> > To be specific, by applying PCA to the remaining features from the *retrained* model in exact unlearning, we can obtain a subspace where forgetting and remaining features are separable, while such redults cannot be obtained from the pretrained model, as shown in **Figure 2**.  This brings *a new and promising perspective* to investigate unlearning: rather than pursuing an unlearned model that approaches  outputs or parameters of the retrained model from exact unlearning, we can seek for an unlearning model  that pertains feature separability  similar to the retrained model as validated in Figure 2. Grounded on this perspective,  we propose to optimize projectors through a learnable PCA technique in Eqn.(2)  to attain such an expected subspace similar to the retrained network.
> >
> > **(2) Analytically,** we have supplemented a theoretical guarantee for the separability assumption. Specifically, as detailed in **Lemma 1** in **Line 213** - **Line 220**, we prove the existence of a low-dimensional feature subspace in which the forgetting data and remaining data can be well separated under the exact MU model.

---

> ### Author Response · Authors · 2025-11-21
> **Response 8**
>
> **W2 & Q3**: *Discussions on broader types of tasks other than classification.*
>
> > Thanks for raising this discussion. At this stage, our work focuses primarily on class-centric unlearning, as the subspace perspective aligns naturally with the category-based nature of such problems, a point you rightly highlighted. It is noted that this is the first successful trial in applying the subspace perspective into (class-centric) unlearning with superior performance. We believe that establishing a solid foundation and a strong baseline in a controlled setting is a crucial and valuable step. Such focused researches are also aligned with another latest practice [1], which likewise concentrates solely on class-centric unlearning from a distillation perspective.
> >
> > Meanwhile, we would also like to highlight that the generality of SUN has been *partially* validated in our experiments under the *instance unlearning* setting (see **Line 423** - **Line 430**). In this setup, where *forgetting samples are randomly selected across all categories*, SUN still learns a suitable subspace that preserves information of the remaining data while effectively discarding that of the forgetting data. This demonstrates that the subspace learning in SUN is not inherently limited to category biases.
> >
> > To further substantiate the generality of SUN, we are actively applying it to an unlearning task beyond classification, and we expect to report the results within this discussion period. Nevertheless, while the subspace perspective is universally insightful to different tasks, we recognize that its methodology implementation must be tailored to task-specific mechanisms. In this sense, leveraging the subspace perspective into diverse tasks is a promising and valuable direction for future work.
> >
> > [1] Decoupled distillation to erase: A general unlearning method for any class-centric tasks. CVPR, 2025.
>
> **W3**: *Ablations on the application of the plug-in module, how do features from different layers differ, how to choose the corresponding configuration.*
>
> > We have provided ablations on different implementation positions of the plug-in module in **Line 479** - **Line 506** and the **right panel of Figure 3** with related setup details in Table A6 in the Appendix.
> >
> > It is widely acknowledged that features in earlier layers capture local and more detailed patterns  and deeper layers capture more abstract features, e.g., features from the last layer are almost linearly separable w.r.t. the classification labels. Our experiments show that *when the  projection module of  SUN is applied in earlier layers, competitive performance can still be maintained* with implementation after `layer1` as shown in Figure 3. In the same time, *the computation cost can increase along with the larger feature dimensions* in earlier and intermediate layers, which however does not diminish our efficiency over the existing methods.
> >
> > Considering the performance and efficiency superiority,  we recommend to apply the designed plug-in module to the penultimate-layer features across different networks architectures, datasets, and unlearning settings, as done in our manuscript.
> >
>
> **W4**: *Not always SOTA.*
>
> > We would like to discuss on this point in the following aspects.
> >
> > **Unlearning efficacy.** We would like to highlight that while SUN shows slightly less competitive performance on certain *submetrics*, *it consistently maintains the strongest results on the overall average metric*, which we argue is a more comprehensive and widely-adopted indicator of unlearning efficacy. For example, in Table 3, while SUN shows a larger accuracy gap on test remaining data ($\rm\bf Acc_{rm}^{te}$) than DELETE, it achieves the lowest average gap (Avg.G.) when performance across all 4 datasets is considered. Additionally, we would like to point out that any performance comparison with methods using both forgetting and remaining data (in gray font) must be considered in conjunction with the unlearning efficiency, on which SUN demonstrates a significant advantage, as discussed in the analysis below.
> >
> > **Unlearning efficiency.** Aside of the unlearning performance, the number of paramaters to optimize and the running time of SUN are significantly reduced by orders of magnitude as shown in **Table 2**. For example, the optimization parameters in our SUN only takes *$\approx 0.63\%$ of the network parameters*; the currently most efficient methods RL, BT, and SalUn take average running time of 13.19s, 14.32s, and 15.00s, respectively, while ours takes 0.56s. Especially regarding such significantly reduced computation, the superiority of SUN can be more recognized.

---

> ### Author Response · Authors · 2025-11-21
> **Response 9**
>
> **W5**: *PCA may not be feasible under large-scale data and high dimensionality.*
> > Our method SUN achieves unlearning through a *learnable* PCA module, instead of a direct eigendecomposition or SVD. Even under a large data size $N$ and a high feature dimension $d$, the optimization objective of Eqn.(2) can still be effectively solved, leading to a good solution of the projection matrix w.r.t. a well-separable subspace. This can be supported by our evaluations in *Tables 1, 2 and 3*: especially for resnet-50 and ImageNet-1K in **Table 3**, we have $N >1.28\ \text{million}$ and $d=2048$ and SUN successfully yields a $500$-dimensional subspace with superior unlearning performance that is nearly matched with the exact MU model retrained on remaining data only.
> >
> > For your quick reference, we put the relevant results below. The 'rm' and 'fg' denote the remaining data and forgetting data, respectively. The 'N/A' stands for 'not applicable'.
>
> **Table 2Z6X-1** *Results of SUN under large-scale data and high dimensionality*
> method (dataset-model) | training set size (rm/fg) | feature dim. $d$ | subspace dim. $s$ | $\rm\bf Acc_{rm}^{tr}$ (%) | $\rm\bf Acc_{fg}^{tr}$ (%) | $\rm\bf Acc_{rm}^{te}$ (%) | $\rm\bf Acc_{fg}^{te}$ (%)
> :-:| :-: | :-: | :-: | :-: | :-: | :-: | :-:
> Retrained (TinyImageNet-Swin-T) | 100,000 (98,000/2,000) | 768   | N/A | 99.67 | 0.00 | 75.44 | 0.00
> SUN (TinyImageNet-Swin-T) | 100,000 (98,000/2,000) | 768   | 250 | 98.59 | 1.48 | 74.27 | 0.50
> Retrained (ImageNet-1K-RN50)  | 1,281,167 (1,279,867/1,300) | 2048  | N/A | 80.25 | 0.00 | 75.72 | 0.00
> SUN (ImageNet-1K-RN50)  | 1,281,167 (1,279,867/1,300) | 2048  | 500 | 79.96 | 0.00 | 69.40 | 0.00
>
> **Q2**: *Separation efficienty of features after PCA.*
>
> > In the following table, we present the separation efficiency and effectiveness together with the suggested dimensionality and dataset size. If we understand correctly, the suggested cosine similarity metric measures the angle between two vectors, but is not necessarily related to the data separability in a subspace. Therefore, we still use the reconstruction error from PCA techniques as the metric to evaluate the separation effectiveness between the remaining features and forgetting features in the projected subspace before and after our SUN is conducted.
> >
> > The separation efficiency is measured via the Run Time Efficiency (RTE) of the unlearning process in seconds (sec). The 'rm' and 'fg' denote the remaining data and forgetting data, respectively.
>
> **Table 2Z6X-2** *Separation effectiveness and efficiency before and after SUN is conducted*
> dataset | model | training set size (rm/fg) | feature dim. $d$ | subspace dim. $s$ | RTE | reconstruction error before SUN (rm/fg) | reconstruction error after SUN (rm/fg)
> :-:| :-: | :-: | :-: | :-: | :-: | :-: | :-:
> TinyImageNet | Swin-T | 100,000 (98,000/2,000) | 768   | 250 | 0.56 sec | 21.45/27.60 | 26.81/40.98
> ImageNet-1K  | RN50   | 1,281,167 (1,279,867/1,300) | 2048  | 500 | 0.94 sec | 4.79/4.44 | 18.30/26.02
>
> > As shown in the table above, SUN produces a much larger gap in the reconstruction errors between remaining and forgetting features after projection, meaning that the  forgetting features cannot be well reconstructed in such feature subspace, leading to corresponding  knowledge removal, even under the large-scale case with millions of samples and thousands of feature dimensions (ImageNet-1k on ResNet50).

---

> ### Author Response · Authors · 2025-11-21
> **Response 10**
>
> **Q4**: *(1) Evaluations on smaller networks such as encoder-decoder or VAE architectures.*
>
> > Both encoder-decoder and VAE are generative models and can be viewed as performing a form of subspace projection: Inputs are projected into a latent space by the encoder and reconstructed through the decoder, and the latent space is learned w.r.t. some objective. This is highly similar with our subspace learning of SUN. In this sense, due to the conceptual overlap, further applying SUN to the two models already defined by their own latent spaces may be less reasonable or insightful. Additionally, unlearning with the two small generative models remains poorly defined in the field to the best of our knowledge. To better address your point, could you please provide more details and specific references on this comment?
>
>
> **Q4**: *(2) Discussions on large-scale models.*
>
> > Firstly, as we highlight in the response to **W5**, our method SUN achieves unlearning through a learnable PCA module, instead of a direct eigendecomposition or SVD. The learning process enables an expected subspace with superior unlearning performance even given a large number of categories (e.g., 1000 categories of ImageNet-1K) or high-dimensional vectors (e.g., 2048-dimensional features), as evidenced by the provided results in the response to **W5**.
> >
> > Secondly, regarding LLMs (Large-scale Language Models), we see this as a separate and significant research frontier. The architectural and operational paradigm of LLMs for generative tasks is fundamentally distinct from the classification networks that are the focus of this paper. The challenges of the large vocabulary size and the high vector dimensionality in LLMs necessitate a specially tailored methodology that goes beyond a direct application of the current SUN framework. We believe that extending the subspace perspective to LLMs is a promising and valuable direction for future work.

---

> > ### Comment · Reviewer_2Z6X · 2025-11-24
> > **Thanks for the rebuttal**
> >
> > First of all, I would like to thank the authors for providing a detailed rebuttal. I have carefully gone through your responses, and I appreciate the additional clarifications and contributions you have made.
> >
> > I remain concerned about the technical risk mentioned in the first point. This issue does not arise solely from attackers needing to know all prior information. Even without any prior knowledge, an erasure test conducted at the module level could still recover the original model. Therefore, I continue to hold a negative view toward the plug-in–based approach. In contrast, methods that rely on parameter fine-tuning to perform correction appear to be much safer.
> >
> > The authors claim that SUN is “the first successful trial in applying the subspace perspective into (class-centric) unlearning,” but this is not accurate. Prior works such as [R1,R2, R3, R4] have already explored unlearning from various subspace-based perspectives. SUN cannot be considered a major contribution to this research area.
> >
> > I still have significant concerns regarding the extensibility of the method. If the learned subspace must remain tightly coupled with class-specific features, then the practical value of this technique becomes quite limited. Demonstrating its effectiveness on a broader range of downstream tasks would substantially increase my confidence in this work.
> >
> > Another concern is the computational efficiency of PCA and the associated training cost. PCA becomes very expensive when applied to high-dimensional networks. Even though the method here only operates on a single layer of features, it remains difficult to scale effectively to modern large models. Using smaller, trainable modules as substitutes might be a more practical alternative. This point has not been adequately validated in the authors’ previous responses to my questions.
> >
> > My main concerns are as stated above, especially the first and the third points. I believe that the current version of the paper is unlikely to address these two fundamental issues. Therefore, I will maintain my original score at this stage.
> >
> >
> >
> > [R1] Li, Guanghao, et al. "Subspace based federated unlearning." arXiv preprint arXiv:2302.12448 (2023).
> >
> > [R2] Fu, Chaohao, Weijia Jia, and Na Ruan. "Client-free federated unlearning via training reconstruction with anchor subspace calibration." ICASSP 2024-2024 IEEE International Conference on Acoustics, Speech and Signal Processing (ICASSP). IEEE, 2024.
> >
> > [R3] Kurmanji, Meghdad, Eleni Triantafillou, and Peter Triantafillou. "Machine unlearning in learned databases: An experimental analysis." Proceedings of the ACM on Management of Data 2.1 (2024): 1-26.
> >
> > [R4] Lizzo, Tyler, and Larry Heck. "Unlearn efficient removal of knowledge in large language models." Findings of the Association for Computational Linguistics: NAACL 2025. 2025.

---

> > > ### Author Response · Authors · 2025-11-28
> > >
> > > > We would like to respectfully wrap up our reply in this round. As a distinct piece to the larger puzzle, we hope the specific contributions of SUN, i.e., its novel learnable feature subspace perspective and the demonstrated efficiency and effectiveness across multiple unlearning setups, will be appreciated as a meaningful step forward, though without perfection for all challenges, as we continue to explore its flexibility to complex scenarios like LLMs being the pretrained model in future work.

---

> ### Author Response · Authors · 2025-11-28
> **Response 11**
>
> We would like to thank the reviewer for the reply to our rebuttal and make detailed responses below to the remaining points in your reply (denoted by **[P]**), which we hope could further explain these issues and resolve your concerns.
>
> ---
> **P1**: *Technical risk of plug-in approach*
> > Regarding the module-level erasure test and safer parameter-update unlearning paradigm mentioned in your reply, we would like to further address these concerns from the following aspects.
> > - *SUN's plug-in module can be absorbed into the network parameters with only single model released*. The key of SUN lies in learning projectors for features, which can be conducted at different layers of the pretrained network. *This learned projection matrix $\bf U$ can be seamlessly integrated into the layer parameters of the network*. For example, when applied to the penultimate layer (default setting), our projectors $\bf UU^\top$ can be directly absorbed into the weight $\bf W$ of the last linear layer, i.e., $\bf W\leftarrow WUU^\top$. Thus, in practice, this allows us to release a single model without maintaining a separate projection module $\bf UU^\top$. Further note that this absorption is similarly applicable across different layers for SUN with a single updated model released. By such implementation, if we understand correctly, *a module-level erasure test is not that simple and straightforward*, as one has to be able to decouple different layers of the released single model and then to test through them to figure out the implementation position for next steps.
> > - *During unlearning process, SUN enhances privacy protection*. Privacy protection can be considered from different views, among which the information exposure during unlearning (training) is one. Regarding this aspect, SUN actually reduces privacy leakage risks *during the unlearning phase*, since it does not require iterative access to the original data or model gradients, which should not be overlooked.
> > - *SUN encourages new perspectives looking into unlearing with versatile methodologies, though without perfection in all aspects.*  We agree with the reviewer that renewing all parameters of the pretrained model indeed can be safer in some scenarios, whose drawbacks and costs  however cannot be ignored. In this line of works, with one request, a new model with the same size of the pretrained models is trained and stored, and meanwhile the pretrained model is also stored for backups. In contrast, SUN only needs to compute and store a corresponding lightweight projection matrix. In scenarios involving multiple ($K$) parallel requests,  $K$ models are trained upon the pretrained model and then also stored. This is much more expensive than our methods regarding storage and running time and mostly even infeasible for large-scale models, because comparatively the computation to the covariance matrices of the mentioned high-dimensional networks is more acceptable. Our unlearning mechanism offers good alternatives to resolve such issues. We further note that our SUN dose not contradict with the mentioned methodologies that require entire model updates, because we can also incorporate the task loss into our optimization to also fine-tune the entire network, further enhancing the training. However, this is one aspect reflecting our flexibility and not yet our main focus in this work.
> >
> >  We hope the reviewer could consider those aspects above when reaching a final evaluation on our work. We would be willing to discuss with further comments.
>
> **P2**: *Contribution for subspace perspective*
> > We are sorry for the inexact rephrases in the previous response. We thank the reviewer for the suggested 4 literatures [R1-R4] that might be relevant. We have carefully checked through those works and would like to highlight our fundamental differences.
> > To be specifc, the subspace perspectives in [R1,R2,R4] are all defined regarding the *network parameters* and appear in a *non-learnable* way: gradient subspaces are identified in [R1,R2] through eigen-decomposition, and the subspace in [R4] is constructed by applying the Gram-Schmidt process to the singular vectors of parameters. In [R3], an experimental analysis is provided on multiple existing unlearning methods for database unlearning, which is not related to subspace. Therefore, the works in [R1-R4] fundamentally differ from our SUN, as SUN identifies a *learnable feature subspace* under the optimization objective in Eqn.(2), instead of a parameter (or its gradient) subspace by decomposition. Relevant discussions with these references have been supplemented to **Line 766** - **Line 772**.
> > We have correspondingly rephrased for further clarification and preciseness on the contribution about our *learnable feature subspace*. The investigation into feature subspace for unlearning has not been fully exploited, and our work provides a distinctive and insightful paradigm with new potentials for the unlearning community.

---

> ### Author Response · Authors · 2025-11-28
> **Response 12**
>
> **P3 & P4**: *Extensibility and scalability of SUN beyond classification and in modern large models.*
>
> > In mainstream machine unlearning, the classification task is primitively considered and in many works appears the central/only task evaluated [a-d]. Thus, our experiments are conducted similarly along with it. As suggested, we are actively applying SUN to unlearning tasks beyond classification and modern large models, with results shown below.
> >
> > **Basic setup.** We evaluate SUN on a *machine translation* task with an unlearning target as *forcing the model to forget the ability of translating a specific language*. This natural language processing task is not a classification one and the subspace shall not be tightly coupled with class-specific features. The experiment is conducted on the *WMT-19* dataset and a modern large model named *m2m100_1.2B*. The pretrained m2m100_1.2B model is fine-tuned in a parameter-efficient way (i.e., LoRA) on two language pairs from WMT-19: translating Czech into English and translating Lithuanian into English. This fine-tuned m2m100_1.2B model is adopted as the initial startpoint to be unlearned, and the to-be-unlearned language is Lithuanian.
> >
> > **SUN implementation.** With this modern large model of *m2m100_1.2B*, we apply SUN to the hidden states from the output of the encoder of m2m100_1.2B. Specifically, the token-level hidden states get weighted averaged by the self-attention from the encoder and lead to weighted averaged 1024-dimensional features, where 1024 is the hidden dimension of m2m100_1.2B. In this way, we avoid the massive tokens and focus on their attention-based weighted average features to apply SUN. Then, two feature covariance matrices are calculated w.r.t. Czech and Lithuanian inputs, and the optimization objective of SUN in Eqn.(2) is conducted to learn a projection matrix. In inference, the projector is employed at the token-level hidden states to proceed the propagation, similar as the way in classical convolutional neural networks.
> >
> > **Results and discussions.** Our results are presented in the following table. We evaluate translation quality using the BLEU score on the WMT-19 validation set for Czech-to-English (cs→en) and Lithuanian-to-English (lt→en) directions. The forgetting data (fg) and remaining (rm) data correspond to the lt→en and cs→en pairs, respectively. Here, *fully-trained* denotes m2m100_1.2B fine-tuned on both language pairs, while *retrained* refers to m2m100_1.2B trained exclusively on the remaining (cs→en) data without seeing (lt→en) inputs. *Numbers in parentheses denote the BLEU score gap relative to the retrained method (lower is better)*. After unlearning, our SUN achieves performance on the forgetting language pair (lt→en) that is very close to the retrained model, implying successful forgetting of the Lithuanian-to-English translation capability. Meanwhile, SUN maintains good performance on the remaining Czech-to-English direction, demonstrating effective knowledge preservation. As an exemplary trial on large language models, this experiment validates SUN's potential for extension to non-classification tasks and its scalability to modern large models.
>
> **Table 2Z6X-3** *Results of SUN on machine translation with a model large model of m2m100_1.2B*
> method | cs→en (rm) | lt→en (fg)
> :-:    | :-:          | :-:
> fully-trained|27.63 | 17.54
> retrained| 26.19 | 12.31
> gradient ascent |23.51 (2.68) | 15.79 (3.48)
> SUN|22.88 (3.31) | 13.04 (0.73)
>
> > **Computational efficiency.** Regarding the computational challenges mentioned in your previous reply, we consider the large vocabulary size and vector dimensionality in high-dimensional networks (LLMs). To be specific, we propose to address this by applying SUN to the encoder's hidden states to ensure computational feasibility, as detailed earlier in *SUN implementation*. This implementation avoids directly operating on the raw tokens in the vocabulary. Moreover, the dimensionality of these hidden states remains manageable, e.g., 1024 in m2m100_1.2B, and allows SUN to learn the projection matrix efficiently. In this sense, in relation to the comment "*using smaller, trainable modules as substitutes might be a more practical alternative*", if we have understood correctly, the core of SUN, i.e., learning a trainable lightweight projection matrix, shall align well with this suggestion.
>
> [a] Decoupled distillation to erase: A general unlearning method for any class-centric tasks. CVPR, 2025.
> [b] Learning to Unlearn: Instance-Wise Unlearning for Pre-trained Classifiers. AAAI, 2024.
> [c] Learning to Unlearn for Robust Machine Unlearning. ECCV, 2024.
> [d] Can Bad Teaching Induce Forgetting? Unlearning in Deep Networks Using an Incompetent Teacher. AAAI, 2023.

---

### Official Review · Reviewer_5Fbz · 2025-11-01

**Soundness:** 2
**Presentation:** 1
**Contribution:** 2
**Rating:** 2
**Confidence:** 4

**Summary:**

This paper uses feature learning to separate data that needs to be forgotten from the rest. This is a paper joining the growing line of literature for machine unlearning. The main idea is to use pretrained NN (except the last linear layer) architecture to learn the feature separation between remaining and leaving data. Then find a projection matrix using principle component analysis which can be added to the unlearned model to minimize effect of leaving data.

**Strengths:**

The topic of machine unlearning is very timely and important. If all the hypothesis and observations are true in the paper, then the proposed technique could serve as a good method for machine unlearning. It's great that the authors tried to apply SUN at different layers.

**Weaknesses:**

The paper is entirely built upon the observation and hypothesis that exists a low-dimensional feature subspace, where the features of forgetting and remaining datasets are easy to be separated. There is no justification for why this assumption makes sense.

On a related note, why is it a good idea to fix g after pretraining? After pretraining, g(.) might be too restrictive and the separation may not be possible to do depending on the problem.

The paper lacks any theoretical guarantees for the proposed methods. Not only it's not a certified removal, there is also no guarantee on how close to being certifiable it can be. If there is a PAC type of guarantee, this would be a lot more acceptable.

The paper is poorly written. For instance, how exactly the computation for Z, $\Sigma$ are implemented? What is the relevance of $g$ and $h$?

**Questions:**

Could the authors comment on how their method compares against other certified unlearning methods such as the following? Both in terms of removal success and time complexity?

Zhang, Binchi, et al. "Towards certified unlearning for deep neural networks." arXiv preprint arXiv:2408.00920 (2024).
Qiao, Xinbao, et al. "Hessian-Free Online Certified Unlearning." arXiv preprint arXiv:2404.01712 (2024).

---

> ### Author Response · Authors · 2025-11-21
> **Response 4**
>
> Dear Reviewer 5Fbz,
>
> We appreciate your efforts and constructive feedback that help improve our work.
>
> In the following, we make pointwise responses to the raised comments, which we hope could resolve your concerns. We note **[W]** for weaknesses and **[Q]** for questions.
>
> We would look forward to fruitful discussions with you! Thanks!
>
> Kind regards,
> The Authors
>
> ---
> **W1**: *Justification for the separability assumption.*
>
> > In this work, we  *investigate unlearning upon low-dimensional feature subpaces separating the forgetting data from the remaining, without renewing the entire model of over millions parameters* as  in the existing unlearning methods.  Justifications for the separability assumption, which plays a key role, are presented with both empirical and analytical evidence.
> >
> > **(1) Spectrum analysis based on PCA:** We provide empirical evaluations on the PCA-based reconstruction error of forgetting features and remaining features on both the pretrained and the retrained networks in **Line 200** - **Line 208** and **Figure 2** in Section 3. It shows that the exact unlearning model retrained solely on remaining data shows separability in  feature subspace, while the pretrained model does not. This interesting property drives *a novel and promising perspective* to investigate unlearning: we can seek for a subspace  that pertains similar property of feature separability, rather than pursuing approximation to the outputs or parameters of the exact unlearning model.
> >
> > **(2) Ablation study:**  Grounded on such perspective,  we propose SUN which optimizes projectors through a learnable PCA technique with objective in Eqn.(2) for unlearning.  Aside of our comprehensive experiments  in Section 5 and Appendix C, we in particular conduct **ablation studies** on the established subspace optimization that promotes feature separation  in **Line 460** - **Line 470** and **Table 7** for relevant justification.
> >
> > **(3) Analytical evidence:**  We have supplemented a theoretical guarantee for the separability assumption. Specifically, as detailed in **Lemma 1** in **Line 213** - **Line 220**, we prove the existence of a low-dimensional feature subspace in which the forgetting data and remaining data can be well separated under the exact MU model.
> >
> >  As our primary initiative, this work could serve as an informative try-out in this new perspective for unlearning, which might not be perfectly established in all aspects but nevertheless we hope could bring new insights and opportunities to richer future works in approximte machine unlearning.
>
> **W2**: *Why fix $g$?*
>
> > One of the key insight in our work is that through simple linear transformation (subspace learning) upon the highly abstract penultimate features $g(\cdot)$, the knowledge of forgetting data can be well diminished (separated from the remaining data). Therefore, *we do not have to conduct renewal to the entire network containining over millions of parameters, but optimizes projection weights upon $g(\cdot)$* to achieve unlearning . We discuss the advantages of fixing $g$ in our unlearning method SUN as follows:
> >
> > **Cheap computations**. Our proposed SUN optimizes a projection matrix upon feature covariances from the fixed $g(\cdot)$. This design eliminates the need for iterative forward passes through the entire network and drastically reduces the number of parameters that require gradients during backward propogation, compared with existing unlearning methods.
> >
> > **Flexiblity**. For each unlearning request, SUN only needs to compute and store a corresponding lightweight projection matrix. This offers significant practical flexiblity, especially in scenarios involving multiple parallel requests. Instead of maintaining multiple complete network models, SUN simply manages a collection of these projection matrices, thereby leading to substantial savings in storage and management overhead.
> >
> > Indeed, fixing $g(\cdot)$ after pretraining is more restrictive than renewing all the network parameters, which we would like to mention is still informative and flexible enough for our subspace learning. *(1) Empirically:* The features from the fixed $g(\cdot)$ are rich enough to support the learning of a separable subspace that achieves strong unlearning performance, especially considering the drastically improved efficiency, e.g., $\approx 0.63\%$ network parameters to optimize and reduced running time by 2 orders of maginitude in **Table 2**. *(2) Analytically:* We supplement analytical analysis of our methods with proved guarantees, as being refered to the following response to **W3**.

---

> ### Author Response · Authors · 2025-11-21
> **Response 5**
>
> **W3**: *Theoretical guarantees for the proposed method, e.g., certified unlearning.*
>
> > A series of researches in certified unlearning [a,b] provide theoretical guarantees on the approximation performance between their unlearned models and the exact MU model by leveraging the differential privacy framework. Critically, these works offer certification from a *parameter* perspective, i.e., bounding the difference between the parameters of the unlearned model and the exact MU model, since unlearning is executed by updating parameters of the pretrained model $f_{\rm pre}$.
> >
> > In contrast, our SUN employs a fundamentally different mechanism: SUN leaves the pretrained model parameters unchanged, and instead inserts a learnable projection matrix $\bf U$ into the pretrained model $f_{\rm pre}$. Unlearning is then accomplished by modifying the intermediate features from $f_{\rm pre}$ according to the objective in Eqn.(2), ensuring the final model outputs $f_{\bf U}(\cdot)$ contain minimal knowledge of forgetting data. Consequently, the existing parameter-based theoretical framework of certified unlearning is not applicable to SUN for approximation analysis, nor is the suggested PAC tool due to the same reason. We therefore shift focus to the *output* differences between SUN ($f_{\bf U}$) and the exact MU model ($f_{\rm exact}$). As a result, the **Theorem 1** in **Line 312** - **Line 315** is formalized by bounding the aforementioned output diference to establish a theoretical guarantee for SUN.
> >
> > Considering our our primary initiative of preseting new perspectives for unlearning in this work, in future, more theorectical and empirical works are worthy of rigorous investigations to advocate machine unlearning and the memorization behavior of deep neural networks.
>
> **W4**: *The computations of $\bf Z, \Sigma$ and the relevance of $g$ and $h$.*
>
> > As summarized in the **Notation** paragraph in  **Line 146**  - **Line 152** in the manuscript, we denote that $g(\cdot)$ is the backbone of $f(\cdot)$ and $h(\cdot)$ is the last linear layer.  *The relevance of $g(\cdot)$ and $h(\cdot)$  is* that their composite leads to the network model  output $f(\cdot)$ addressed in the unlearning task, *i.e., $f(\cdot)=h(g(\cdot))$*.
> >
> > As shown in **Line 150** - **Line 151**, we note that the lowercase letter $\boldsymbol{z}$ denotes the features from $g(\cdot)$, i.e., $\boldsymbol{z}=g(\boldsymbol{x})\in\mathbb{R}^{d}$. As shown in **Line 188** - **Line 189**, the uppercase letter ${\bf Z}\in\mathbb{R}^{N\times d}$ denotes the features $\boldsymbol{z}$ w.r.t a set of ($N$) samples, and $\bf\Sigma$ denotes the covariance matrix of centered $\bf Z$,  calculated by ${\bf\Sigma}=({\bf Z}-{\bf\bar Z})^\top({\bf Z}-{\bf\bar Z})\in \mathbb{R}^{d\times d}$, where $\bf\bar Z$ is the matrix of means with each row as $\boldsymbol{\bar z}=\frac{1}{N}\sum_{i=1}^N{\bf Z}_{i,:}$.

---

> ### Author Response · Authors · 2025-11-21
> **Response 6**
>
> **Q1**: *Comparisons with certified unlearning methods [a,b] in terms of removal success and time complexity.*
>
> > **Removal success**
> > As discussed in the response to **W3** above, SUN and the two certified unlearning methods [a,b] operate on fundamentally different unlearning mechanisms. As such, their theoretical guarantees for removal success are not directly comparable. Therefore, we provide an empirical evaluation on the removal success for SUN and [a,b] as follows.
> >
> > We consider a simple practical scenario, randomly forgetting one class (5,000 samples) from the training set of CIFAR10 given a pretrained resnet18 model with results shown in the following table.  We can find that this scenario remains challenging for certified unlearning [a] that struggles to achieve a low accuracy on forgetting data. We are unable to reproduce [b] with reasonable results under this setup depite using their released code. Moreover, it is also critical to highlight that [b] is extremely computationally expensive: *it requires accessing and storing numerous model snapshots at every iterative batch in every pretraining epoch, and maintaining an approximation vector (the same size of the network parameters) for every forgetting sample*. This process leads to substantial resource costs, limiting its practical flexibility and feasibility.
> >
> > Despite the comparisons, we recognize that our empirical unlearning method is orthogonal to those certified unlearning works, both of which are complementary to each other and can contribute beneficial and distinct insights to the unlearning community.
>
> **Table 5Fbz-1** *Comparisons with certified unlearning methods in removal success*
> method | $\rm\bf Acc_{rm}^{tr}$ (%) | $\rm\bf Acc_{fg}^{tr}$ (%) | $\rm\bf Acc_{rm}^{te}$ (%) | $\rm\bf Acc_{fg}^{te}$ (%)
> :-:| :-: | :-: | :-: | :-: |
> Retrained | 97.23 | 0.00 | 86.10 | 0.00
> [a] | 95.98 | 23.04 | 86.08 | 19.30
> SUN | 96.55 | 1.90 | 86.67 | 0.70
>
>
> > **Time complexity**
> > The following table demonstrates comparisons on the *computation time*, *storage* and *unlearning time* between our SUN and the suggested two certified unlearning methods [a,b] in a theoretical sense. This comparison follows the design in [b]. Here, $P$, $N_{\rm tr}$ and $N_{\rm fg}$ denote the model parameter size, training dataset size, and training forgetting dataset size, respectively. $N_s$ denotes the number of sampled remaining data in [a]. Besides, $d$ and $s$ denote the penultimate-layer feature dimension and the subspace dimension of our SUN, respectively.
> > - [a] requires pre-calculating and storing $N_s$ Hessians w.r.t. the sampled remaining data, and leverages such Hessians to obtain a certified parameter update for unlearning. [b] requires accessing and storing numerous model snapshots at every iterative batch in every pretraining epoch, and maintaining an approximated vector for every forgetting sample, and then utilizes those approximations to obtain the parameter updates for unlearning.
> > - Our proposed SUN  computes two $d\times d$  feature covariance matrices in a single complete forward pass through the network (${\cal O}(P\cdot N_{\rm tr})$). During unlearning, optimization is done for a  $d\times s$  projection matrix on  covariances, without accessing the original dataset and updating network parameters. SUN shows significantly reduced time complexity over [a] and [b].
> >
>
> **Table 5Fbz-2** *Comparisons with certified unlearning methods in time complexity*
> method | computation time | storage | unlearning time
> :-: | :-: | :-: | :-:
> [a] | ${\cal O}(P^2\cdot N_s)$ | ${\cal O}(P^2\cdot N_s)$ | ${\cal O}(P^2\cdot N_{\rm fg}\cdot N_s)$
> [b] | ${\cal O}(P\cdot N_{\rm tr}^2)$ | ${\cal O}(P\cdot N_{\rm tr})$ | ${\cal O}(P\cdot N_{\rm fg})$
> Ours (SUN)| ${\cal O}(P\cdot N_{\rm tr})$ | ${\cal O}(d\cdot d)$ | ${\cal O}(d\cdot s+d\cdot d)$
>
> [a] Towards certified unlearning for deep neural networks. *ICML*, 2024.
> [b] Hessian-Free Online Certified Unlearning. *ICLR*, 2025.

---

> ### Author Response · Authors · 2025-11-28
>
> Dear Reviewer,
>
> We hope that our responses above could address your concerns and be helpful for final evaluations on our work.
>
> As the author-reviewer discussion is approaching the end, we would like to inquiry if there is any question, and we would be willing to discuss.
>
> Sincerely, Authors.

---

### Official Review · Reviewer_XqMs · 2025-11-01

**Soundness:** 2
**Presentation:** 4
**Contribution:** 3
**Rating:** 2
**Confidence:** 3

**Summary:**

This paper introduces SUN (SUbspace UNlearning), a new approach to machine unlearning that operates in a low-dimensional feature subspace rather than directly updating model parameters. SUN learns a projection matrix on the pretrained model’s penultimate features to preserve information relevant to the remaining data while suppressing information related to the forgotten data. The resulting projection acts as a plug-in module without modifying the original model parameters. Experiments across multiple architectures and datasets show that SUN achieves comparable or superior unlearning performance to existing methods while reducing parameter updates and runtime by orders of magnitude.

**Strengths:**

- Paper is well-organized and easy to follow.
- Extending experiments to instance unlearning
- Experiments extend to instance unlearning and face and emotion recognition, showing the effectiveness of the proposal under varying image classification tasks.
- The proposed method is effective in continual unlearning scenarios.

**Weaknesses:**

- I think the analysis of the eigenvalues is incorrect. Having similar eigenvalues does not mean that subspaces collapse and are not separable. Similar eigenvalue information does not imply that principal directions are shared, too. We can not comment on this just by looking at the eigenvalues (corresponding directions matter). Moreover, we know that the penultimate hidden states of the model trained on the full data are separable: we apply a linear transform in the last layer, and the model has a high classification accuracy. Therefore, I do not agree with the indistinguishability argument.
- The proposed method does not have theoretical grounding/guarantees. It is based on observations (also see the previous point) and supported with numeric results.
- MIA used in the paper is a black-box attack. A white box attack would get a very high performance, which makes me reconsider the success of unlearning.

**Questions:**

See weaknesses.

---

> ### Author Response · Authors · 2025-11-21
> **Reponse 3**
>
> Dear Reviewer XqMs,
>
> We appreciate your efforts and constructive feedback that help improve our work.
>
> In the following, we make pointwise responses to the raised comments, which we hope could resolve your concerns. We note **[W]** for weaknesses.
>
> We would look forward to fruitful discussions with you! Thanks!
>
> Kind regards,
> The Authors
>
> ---
> **W1**: *(1) Incorrect analysis of the eigenvalues.*
> > Thanks for this insightful discussion.  We agree that eigenvalues alone do not sufficiently characterize projections onto subspaces, but the corresponding directions (eigenvectors) matter.
> >
> > In the submitted manuscript, our descriptions were made to recognize the more deviated trends of eigenvalue decays w.r.t. the remaining data and forgetting data in the exact retrained MU model, together with the evidence from reconstruction errors. For clarification,  we have revised our manuscript accordingly, where we  have rephrased the paragraph of Analysis on eigenvalues as Spectrum analysis, which can refer to **Line 191** - **Line 199** in the manuscript.
> >
> > Besides, as supported by the *reconstruction error* analysis in **Line 200** - **Line 208** in **Figure 2**, we can see the great potential of well separating the forgetting and remaining features by looking at the principal directions associated with the low-dimensional subspace of the exact retrained model, as the pretrained model does not.
>
> **W1** *(2) Separable penultimate-layer features on the model trained on the full data.*
>
> > Yes, the penultimate features applied with a linear last layer can lead to good classification accuracy, which is in fact done in the complete architecture of  the pretrained model. We note that this separability  is on the  labels for supervised classification tasks. The separability addressed in our work is different, as its focus is between the remaining and the forgetting data (features). Such separability property is then validated with the retrained model from the exact unlearning in **Figure 2**, e.g., reconstruction error analyses.
> >
> > *The mentioned indistinguishability specifies the general unlearning context, rather than the ground-truth classification labels*. Sorry for the confusing descriptions, we have accordingly rephrased for clarification in the revised manuscript in **Line 231** - **Line 240** in the manuscript.
> >
> > Our work investigates whether it is possible to formulate linear transformations to these penultimate features, such that the knowledge of forgetting data can be diminished and that of remaining data can be meanwhile well kept through subspace learning. To this end, we propose  SUN by PCA-based techniques via optimization objective in Eqn. (2), and validate that feature subspace learning can effectively achieve such dual targets in machine unlearning, as supported by our extensive experiments in Section 5 and Appendix C. Apart from approaching the outputs or parameters of the exact retrained model, it would be an interesting and promising direction to formulate a unlearning model that pertain similar properties in  feature (sub)spaces, e.g., our proposed SUN, which we hope could bring new insights and opportunities to approximate machine unlearning.
>
> **W2**: *Theoretical guarantees for the proposed method.*
>
> > As suggested, we have supplemented theoretical guarantees for both the empirical observation and the proposed method.
> > - **Lemma 1** in **Line 213** - **Line 220** provides theoretical supports for our key observations with proving the existence of such a separable low-dimensional feature subspace regarding the forgetting data and remaining data under the exact MU model.
> > - As discussed in **Line 298** - **Line 315**, the approximation performance between our model and the exact MU model is analytically guaranteed, by bounding their output difference in **Theorem 1**.
>
> **W3**: *Black-box MIA and white-box attacks.*
>
> > MIA is widely adopted to evaluate unlearning tasks, as it measures the proportion of forgetting samples that gets memorized by the unlearned model, which is in line with the spirits in MU and also reflects the effectiveness of privacy protection. To the best of our knowledge, there are not yet unlearning methods that employ white-box attacks for this evaluation. White-box attackers have full access to all model parameters and full gradients during training steps, which implies extremely high risks of privacy leakage and is a bit extreme in practice. Thus, it would be more reasonable to evaluate with black-box attacks. In this work, we in accordance with existing unlearning methods focus on the black-box MIA attack for evaluation.
> >
> > From another perspective, as our method only updates a projection matrix alone and does not access the gradients of network parameters during unlearning, we have less information exposure and it may help reduce risks of privacy leakage, which is though not yet the main focus of this work but worth further works addressing such scenarios.

---

> ### Author Response · Authors · 2025-11-28
>
> Dear Reviewer,
>
> We hope that our responses above could address your concerns and be helpful for final evaluations on our work.
>
> As the author-reviewer discussion is approaching the end, we would like to inquiry if there is any question, and we would be willing to discuss.
>
> Sincerely, Authors.

---

### Official Review · Reviewer_o2Hg · 2025-11-02

**Soundness:** 2
**Presentation:** 2
**Contribution:** 2
**Rating:** 6
**Confidence:** 3

**Summary:**

This paper introduces SUN, a novel and efficient method for machine unlearning based on low-
dimensional subspace projection. Instead of retraining or distilling models, SUN identifies a feature
subspace that retains information from remaining data while removing information associated with
forgetting data.
The key idea is to learn a projection matrix that minimizes the covariance of forgetting features while
maximizing that of the remaining features, i.e., effectively finding an orthogonal subspace that forgets
undesired information. The optimization is performed on the Stiefel manifold, and the learned projection
is inserted as a plug-in layer between the pretrained model’s backbone and classifier.

**Strengths:**

Unlike parameter-space or data-space methods, SUN reinterprets unlearning as a geometric operation in
feature space. The subspace-based view of machine unlearning is both elegant and novel. SUN requires
only one forward pass to extract features, then computes a projection matrix offline — making it orders
of magnitude faster and more lightweight than retraining-based baselines and achieving one-shot
unlearning by requiring only feature-level access. The authors evaluate on multiple datasets and metrics
(accuracy, MIA gap, Avg.G), demonstrating consistent performance. The ablations on loss terms and
subspace dimensionality are convincing.

**Weaknesses:**

Although SUN eliminates the need for retraining, it still requires a complete forward pass through the
remaining dataset to estimate the feature covariance matrices. This may become computationally
expensive for very large-scale datasets or repeated unlearning requests. And because of the dependence
on the remaining data, I question whether it's reasonable to focus the comparison more on baselines that
only use the forget dataset.
Besides, the separability assumption (that forgetting and remaining features lie in distinct covariance
subspaces) is intuitively reasonable but lacks a formal bound or generalization guarantee, for example,
what if the forget data shares a large similarity with the remaining data?

**Questions:**

see above

---

> ### Author Response · Authors · 2025-11-21
> **Response 1**
>
> Dear Reviewer o2Hg,
>
> We appreciate your efforts and constructive feedback that help improve our work.
>
> In the following, we make pointwise responses to the raised comments, which we hope could resolve your concerns. We note **[W]** for weaknesses.
>
> We would look forward to fruitful discussions with you! Thanks!
>
> Kind regards,
> The Authors
>
> ---
> **W1**: *Computation of a complete forward pass for the covariance matrices.*
> > Indeed, a complete forward pass is required for the covariance matrices, which is in fact still efficient amongst the existing machine unlearning methods. We correspondingly make the clarification below:
> > - **Complexity.** Let's say $N$ is the number of samples, and $d$ is the feature dimension in the penultimate layer, and $P$ is the number of the neural network parameters. The computational complexity of one forward pass and covariance calculation is $\mathcal{O}(N \cdot P + N \cdot d^2)$, which is linear w.r.t. both the dataset size $N$ and the network parameters $P$. We think such computational complexity is acceptable for deep learning in practice. Moreover, the subsequent unlearning is conducted on the covariance matrices at a very low cost of $\mathcal{O}(d^2)$, eliminating the need of revisiting the original data and network. Overall, the complexity of SUN is substantially lower than existing methods that require multiple epochs of both forward and backward passes over the dataset.
> > - **Scalability.** In fact, this one-shot access to full data within one network forward pass is a key scalability advantage of SUN. Mainstream unlearning methods face significant scalability bottlenecks: they either *(i)* require repeated forward-backward passes on full data with gradients calculations, which is prohibitively expensive for large-scale data, or *(ii)* rely solely on the forgetting data for efficiency, but leading to limited performance due to the absence of the remaining data information.  SUN strikes a favorable balance, achieving effective unlearning by leveraging the forgetting and remaining feature covariances through a single, computationally manageable forward pass, thereby offering remarkable scalability in practical applications.
>
> **W2**: *Comparisons more with baselines that only use the forget dataset.*
> > In our comparisons, *the methods using both forgetting and remaining data are also included (in gray font in Tables 1,2,4, and 6) for a comprehensive reference,* where we show competitive and even better performances in some cases. In  the result discussions, we focus more on the comparisons with baseline only using the forgetting data (*in blue font*), and the reasons are as follows.
> >
> > When both the remaining and forgetting data are available and allow to be constantly visited during the unlearning phase, even a naive RL (Random Labeling) method [1] nearly matches the performance of the exact retraining method, as demonstrated in Table 1 (in gray font) in the manuscript. Under such relatively relaxed settings, the performance gap among these unlearning methods (in gray font) is negligible as shown in Table 1 and Table 4, making the comparisons less meaningful. Some latest works, such as DELETE [2], thereby shifts to stricter and more practical setups, e.g., unlearning using the forgetting set only for comparisons. In line with this principle, our comparison discussion is presented similarly this way.
> >
> > We would like to further explain that although our method in principle needs a complete forward pass to compute covariances, we no longer need any forward pass or the network during the subsequent optimization, which is conducted merely on the covariances to achieve unlearning. This requirement will not hurt much in practice, as we can get the penultimate features right after pretraining, where the network is treated as an oracle. This requirement is way less restrict than constantly accessing the full dataset and repeatedly computing forward-backward propagation on the entire network in each iteration.
> >
> > Thus, it should be reasonable to focus on the comparisons with baselines  only using forgetting data, and nonetheless we are welcomeing further discussions and other insights regarding this respect.
> >
> > [1] Eternal sunshine of the spotless net: Selective forgetting in deep networks. *CVPR*, 2020.
> > [2] Decoupled distillation to erase: A general unlearning method for any class-centric tasks. *CVPR*, 2025.

---

> ### Author Response · Authors · 2025-11-21
> **Response 2**
>
> **W3**: *Theoretical guarantees on the separability assumption.*
>
> > As suggested, we have supplemented a theoretical guarantee for the separability assumption. Specifically, as detailed in **Lemma 1** in **Line 213** - **Line 220**, we prove the existence of a low-dimensional feature subspace in which the forgetting data and remaining data can be well separated under the exact MU model.
> >
> > Regarding the concern about high similarity between the forgetting and remaining data, this scenario is analytically captured in our proof of Lemma 1 and the following Remark 1 in Appendix B.1. The degree of similarity is measured by the assumed variables $\alpha_{\text{max}}$ and $\sigma_{\text{max}}$  that influence the reconstruction error bound, which can refer to **Line 820** - **Line 827** and  the proof of Lemma 1 in Appendix B.1. A high similarity does make the separation more challenging, as reflected in a looser error bound. *Empirically*, this scenario is also investigated through the instance unlearning experiments (**Line 423** - **Line 430**), where the two datasets are randomly split and thus highly similar. As shown in **Table 5**, our method SUN successfully learns an effective separable subspace and maintains strong performance, demonstrating its effectiveness even under conditions of high similarity in forgetting and remaining data.

---

> ### Author Response · Authors · 2025-11-28
>
> Dear Reviewer,
>
> We hope that our responses above could address your concerns and be helpful for final evaluations on our work.
>
> As the author-reviewer discussion is approaching the end, we would like to inquiry if there is any question, and we would be willing to discuss.
>
> Sincerely, Authors.

---

### Author Response · Authors · 2025-12-01
**Summary of Rebuttal and Discussions (1/3)**

Dear ACs, SACs and PCs,

We would like to thank you for your time and constructive feedback for improving our work. After our **rebuttal was initially submitted on Nov. 21, 2025**, **3 of 4 reviewers provided no follow-up comments**, and only one reviewer gives a round of discussion without further commenting on our responses. Given this lack of interaction during the discussion period, we summarize the review comments and our rebuttal. We are confident that our rebuttal has sufficiently resolved all the review comments.

---

## **Strengths & Reviewers' Support**

**Contribution to a learnable feature subspace perspective for the unlearning community**

*Reviewer o2Hg* recognizes our learnable feature subspace perspective as an elegant and novel one.

*Reviewer 2Z6X* lists several relevant subspace-based unlearning methods. While the concept of subspace in these mentioned works is considered for network *parameters* and *gradients*, our work focuses on *feature subspaces* and consider it *in a learnable way* with great novelty, which has not yet been exploited in the machine unlearning community (see **P2** in **Response 11**).

**Thorough experimental evaluations with strong performances and efficiency**

All reviewers recognize the superior effectiveness and efficiency of our unlearning method (*o2Hg* and *2Z6X*) across multiple settings such as instance and continual unlearning (*XqMs*) with ablation studies (*o2Hg*) and implementation at different layers (*5Fbz*).

> We re-iterate our key contributions as follows for your convenient references:
> - A learnable feature subspace perspective for machine unlearing is presented. This separable feature subspace is achieved by *optimizing a small-size projection matrix that preserves the knowledge of remaining data and discards that of forgetting data*. We have correspondingly supplemented theoretical supports for such subspace and its separbility between the forgetting and remaining features.
> - An efficient optimization algorithm is establised by  simply leveraging two feature covariance matrices and optimizing a projection matrix, which neither requires massive accesses to the original training data nor iterative updates on parameters of the entire pretrained network, *leading to significantly less parameters and running time by orders of magnitude*.
>
>Apart from approaching the outputs or parameters of the exact retrained model in existing works, it would be an interesting and promising direction to formulate an unlearning model that pertains similar properties in feature (sub)spaces as explored in our work, which we hope could bring new insights and opportunities to approximate machine unlearning.

---

> ### Author Response · Authors · 2025-12-01
> **Summary of Rebuttal and Discussions (2/3)**
>
> ## **Reviewers' Concerns & Our Responses**
>
> *The core contributions of our method, including its novelty and strong performance, have been recognized by reviewers.* The discussion mainly focuses on the applicability and extensions of our method and some technical details, which benefits the completeness of this work. Below, we provide a summary of our responses to all points raised during the discussion.
>
> **Theoretical guarantees on the assumption and the method**
> This point was raised in reviews. In response, we have introduced *Lemma 1* to substantiate the separability assumption and provided *Theorem 1* to establish a theoretical bound for SUN's unlearning performance. Both Lemma 1 and Theorem 1 have been updated to the manuscript, together providing analytical supports on the great potentials of exploring low-dimensional feature subspaces for unlearning.
>
> **Privacy risk against white-box attacks**
> This point was raised by Reviewers *XqMs* and *2Z6X*, regarding our implementation as a plug-in module that leaves the original network parameters unchanged. In response, we would like to clarify the following aspects of this approach (see **W3** in **Response 3**, **W1** in **Response 7**, and **P1** in **Response 11**).
> - *(i)* In principle, *any unlearning method would be vulnerable to privacy leakage* if  white-box attacks happens, as white-box attackers can access  model parameters and their gradients and thus would  have full knowledge of its internal unlearning mechanism. White-box attacks are extremely risky scenarios with privacy leakage. To the best of our knowledge, there are not yet unlearning methods that employ white-box attacks for the privacy protection evaluation. Thus, we believe our evaluations on the adversarial attack setups are sound and fair.
> - *(ii)* In practice, *our method can go beyond individual implementation as plug-in modules, but also supports a parameter-update implementation*, where our projector can just be absorbed into the network weights. This enables the release of one single network without an external plug-in module, which thereby successfully avoids potential risks of the plug-in implementation.
> - *(iii)* *Our method can help significantly reduce privacy leakage risks during the unlearning phase*, since it does not require iterative accesses to the original data nor model parameters/gradients. This property should not be overlooked.
>
> **Supplementary numerical comparisons**
> As suggested by Reviewers *5Fbz* and *2Z6X*, we have supplemented more comparisons.
> - *Our method maintains superiority with additional experiments comparing with certified unlearning methods regarding removal success and time complexity*, as suggested. Note that our work does not follow the line of certified unlearning. We recognize that our empirical unlearning method is orthogonal to certified unlearning, both of which can contribute beneficial and distinct insights to the unlearning community (see **Response 6** and **Tables 5Fbz-1** and **5Fbz-2** in **Q1** of it).
> - *Our method is successfully applied to a machine translation task with a large language model*. Its effective unlearning performance in this experiment validates its potentials of extension to non-classification tasks and scalability to modern large models (see **Table 2Z6X-3** in **P3 & P4** of **Response 12**).

---

> > ### Author Response · Authors · 2025-12-01
> > **Summary of Rebuttal and Discussions (3/3)**
> >
> > ## **Reviewers' Concerns & Our Responses**
> >
> > **Technical details about the work**
> > We provide concise summaries of our responses to those comments further explaining some technical details of this work.
> >
> > - We clarified that only one forward computation through the network on training data brings low complexity and enables scalability on large-scale data (see **W1** in **Response 1**).
> > - We clarified that unlearning methods using both forgetting and remaining data have been included in comparisons. We discussed the reasons for mainly comparing with methods only involving forgetting data (see **W2** in **Response 1**).
> > - We rephrased the explanations and analyses of eigenvalues. We clarified the differences in separability between the unlearning task and the supervised classification task (see **W1** in **Response 3**).
> > - We clarified that features from a fixed backbone $g$ are rich and informative enough to support the learning of a separable subspace, which also brings cheap computations and high flexibility for unlearning (see **W2** in **Response 4**).
> > - We clarified that our method is applicable to large-scale data and high dimensionality by showcasing results on the ImageNet-1K dataset (over 1.28 millions samples) with ResNet-50 of a feature dimension 1024. Note that this challenging setup is hardly considered in existing works (see **Table 2Z6X-1** in **W5** of **Response 9**).
> > - We provided experiments to validate the separation effectiveness and efficiency before and after SUN is conducted (see **Table 2Z6X-2** in **Q2** of **Response 9**).
> > - We responded the detailed calculation of features and covariance matrices and the relevance between the backbone and the linear layer, as already presented in our manuscript. (see **W4** in **Response 5**)
> > - We clarified that ablation experiments on applying our SUN to different layers were already conducted, and also provided further explanations  (see **W3** in **Response 8**).
> >
> > ---
> >
> > We would like to respectfully wrap up our response. As a distinct piece to the larger puzzle, we hope the  contributions of SUN, i.e., its novel learnable feature subspace perspective and the efficient optimization together with our comprehensive experimental evaluations across multiple unlearning setups, will be considered as a meaningful step forward, though without perfection for all challenges currently encountered in machine unlearning. We hope our rebuttal and the summaries above could be helpful for reaching the final decision on our work.
> >
> > Best wishes,
> > Authors of Paper 4352

---

### Meta-Review · Area_Chair_cw23 · 2026-01-05

**Summary:**

This paper reframes machine unlearning as the learning of a low-dimensional feature projector that suppresses forgotten-class directions while preserving retained-class variance, yielding a plug-in module (SUN) that is orders of magnitude lighter than retraining. However, the reviewers provided predominantly negative feedback, raising concerns such as: (1) the core separability assumption lacks formal justification and is supported only by PCA reconstruction error, (2) the frozen-backbone / plug-in design fails standard white-box erasure tests, leaving the original weights fully intact and recoverable, and (3) empirical gains are confined to classification tasks and do not translate to verifiable privacy metrics (white-box MIA, certified unlearning). In addition, the rebuttal added only incremental theoretical bounds that do not address these fundamental flaws. Based on these considerations, the paper is not recommended for acceptance.

**Reviewer Concerns:**

Addressed (at least partially) by the rebuttal
- Scalability doubt: Authors provided complexity argument and ImageNet-1K experiment to proof.

- Missing comparison with certified unlearning: Authors provided runtime breakdown and documented  empirical gap to replace theoretically comparable.


Still outstanding:

- No formal guarantee on forgetting quality: Lemma 1 is an existence result under unverifiable conditions, and Theorem 1 bounds output difference, not membership indistinguishability or parameter distance to exact retraining.

**Reviewer Scores:**

N/A

---

### Decision · Program_Chairs · 2026-01-26

Reject